



Natural Hazards
and Earth System
Sciences

# Stochastic system dynamics modelling for climate change water scarcity assessment of a reservoir in the Italian Alps

**Stefano Terzi**[1,2,3], **Janez Sušnik**[4], **Stefan Schneiderbauer**[1,3,5], **Silvia Torresan**[2,6], and **Andrea Critto**[2,6]

[1]Institute for Earth Observation, Eurac Research, Viale Druso 1, 39100, Bolzano, Italy
[2]Department of Environmental Sciences, Informatics and Statistics, Ca' Foscari University of Venice,
Via Torino 155, 30172, Venice, Italy
[3]Institute for Environment and Human Security (UNU-EHS), United Nations University,
Platz der Vereinten Nationen 1, 53113 Bonn, Germany
[4]Land & Water Management, IHE Delft Institute for Water Education, 2601DA, Delft, the Netherlands
[5]Department of Geography, University of the Free State, Bloemfontein, South Africa
[6]Fondazione Centro Euro-Mediterraneo sui Cambiamenti Climatici (CMCC),
via Augusto Imperatore 16, 73100, Lecce, Italy

**Correspondence:** Stefano Terzi (stefano.terzi@eurac.edu)

**Abstract.** Water management in mountain regions is facing multiple pressures due to climate change and anthropogenic activities. This is particularly relevant for mountain areas where water abundance in the past allowed for many anthropogenic activities, exposing them to future water scarcity. Here stochastic system dynamics modelling (SDM) was implemented to explore water scarcity conditions affecting the stored water and turbined outflows in the Santa Giustina (S. Giustina) reservoir (Autonomous Province of Trento, Italy). The analysis relies on a model chain integrating outputs from climate change simulations into a hydrological model, the output of which was used to test and select statistical models in an SDM for replicating turbined water and stored volume within the S. Giustina dam reservoir. The study aims at simulating future conditions of the S. Giustina reservoir in terms of outflow and volume as well as implementing a set of metrics to analyse volume extreme conditions.

Average results on 30-year slices of simulations show that even under the short-term RCP4.5 scenario (2021–2050) future reductions for stored volume and turbined outflow are expected to be severe compared to the 14-year baseline (1999–2004 and 2009–2016; −24.9 % of turbined outflow and −19.9 % of stored volume). Similar reductions are expected also for the long-term RCP8.5 scenario (2041–2070;

−26.2 % of turbined outflow and −20.8 % of stored volume), mainly driven by the projected precipitations having a similar but lower trend especially in the last part of the 2041–2070 period. At a monthly level, stored volume and turbined outflow are expected to increase for December to March (outflow only), January to April (volume only) depending on scenarios and up to +32.5 % of stored volume in March for RCP8.5 for 2021–2050. Reductions are persistently occurring for the rest of the year from April to November for turbined outflows (down to −56.3 % in August) and from May to December for stored volume (down to −44.1 % in June). Metrics of frequency, duration and severity of future stored volume values suggest a general increase in terms of low volume below the 10th and 20th percentiles and a decrease of high-volume conditions above the 80th and 90th percentiles. These results point at higher percentage increases in frequency and severity for values below the 10th percentile, while volume values below the 20th percentile are expected to last longer. Above the 90th percentile, values are expected to be less frequent than baseline conditions, while showing smaller severity reductions compared to values above the 80th percentile. These results call for the adoption of adaptation strategies focusing on water demand reductions. Months of expected increases in water availability should be considered periods for water accumulation while preparing for

potential persistent reductions of stored water and turbined outflows. This study provides results and methodological insights that can be used for future SDM upscaling to integrate different strategic mountain socio-economic sectors (e.g. hydropower, agriculture and tourism) and prepare for potential multi-risk conditions.

## 1 Introduction

Mountains serve as "water towers" providing freshwater to a large portion of the global population (IPCC, 2014, 2018; Kohler et al., 2014; United Nations, 2012; Viviroli et al., 2007). Climate change affects mountain environments more rapidly than many other places, with impacts on glaciers, snow precipitation, water flows and the overall supply of water (Viviroli et al., 2011; Barnett et al., 2005). These impacts call for the need to shift water management towards more sustainable and adaptive practices. Adaptation delays and unpreparedness to water availability changes can spread consequences across multiple systems, from natural ecosystems to anthropogenic activities relying on water (van den Heuvel et al., 2020; Mehran et al., 2017a; Fuhrer et al., 2014; Xu et al., 2009).

The European Alps are among those mountain regions where water abundance in the past allowed for the development of activities with intensive water use such as large hydropower plants and irrigated agriculture, making them susceptible to future impacts regarding reduced water availability (Beniston and Stoffel, 2014; Majone et al., 2016; Permanent Secretariat of the Alpine Convention, 2009). That is, in many Alpine regions the socio-ecological systems are unprepared for water scarcity, and hence the impacts of water shortage can be more severe (Di Baldassarre et al., 2018).

Previous studies have assessed the hydrological processes involved in mountain environments, looking at the overall hydrological dynamics (Bellin et al., 2016) or specifically assessing topics such as glacier melt and runoff (Huss and Hock, 2018; Farinotti et al., 2012) and snowpack dynamics (Etter et al., 2017; Wever et al., 2017). However, the interplay connecting natural processes and socio-economic activities calls for further research. There is a need to implement methodologies with the ability to unravel this complexity in order to effectively tackle climate-related water issues.

System dynamics modelling (SDM) is a methodology used to improve the understanding of complex systems and their dynamic interactions. It makes use of four main modelling elements: (i) stocks (system state variables) – "accumulating" material (e.g. water in a reservoir), (ii) flows (variable's rate of change) – moving material into and out of stocks (e.g. river inflows and outflows), (iii) converters – parameters influencing the flow rates (e.g. temperature variable acting to alter evaporation from a water body), and (iv) connectors – as arrows transferring information within

the model (e.g. linking the monthly effects on reservoir water discharges; Sterman et al., 2000). The combination of these elements is applied to represent temporal changes in system elements accounting for endogenous and exogenous influences on system behaviour. This concept encourages a systems thinking approach, splitting large systems into subsystems and progressively increasing their interactions and complexity (Gohari et al., 2017; Mereu et al., 2016). SDMs can combine different metrics and indices, improving models by adding social, economic and environmental sectors (Terzi et al., 2019). Moreover, SDM can implement a graphical interface, supporting the visualization of interactions and feedback loops during participatory approaches.

While SDM was developed to improve industrial business processes (Forrester, 1971), it has been successfully applied to model human and natural resources interactions (Meadows et al., 2018). Moreover, SDM applications span a wide range of problems, spanning climate change risk assessments (Duran-Encalada et al., 2017; Masia et al., 2018), water management issues (Davies and Simonovic, 2011; Gohari et al., 2017), disasters studies (Menk et al., 2020; Simonovic, 2001, 2015), water–energy–food nexus studies (Sušnik et al., 2018; Davies and Simonovic, 2008) and applications fostering participatory modelling (Malard et al., 2017; Stave, 2010). SDM has therefore been proved to be a useful tool to study complex interactions and dynamic behaviour in a wide variety of complex systems (Ford, 2010).

However, SDM also shows some limitations, such as (i) the limited spatial representation, since it works with lumped regions, although recent research has coupled SDM to GIS (geographic information system) to account for spatially explicit system dynamics (Neuwirth et al., 2015; Xu et al., 2016); (ii) the ease of creating complex what-if scenarios that can be difficult to validate but which can be useful to explore systems behaviour under potential futures; and (iii) the fact that applications usually account for deterministic expert-based assumptions on existence and the type of variables' interactions (Mereu et al., 2016; Sahin and Mohamed, 2014; Sušnik et al., 2013), although statistical analysis of trends and interactions are crucial under uncertain climate change and risk assessments and recent analyses have been used for probabilistic SDM output (Sušnik et al., 2018).

These limitations call for methodological improvements. The combination of statistical methods with SDM represent a valuable integration to overcome the current limitations involved in deterministic expert-based assumptions of variables' interactions and dependencies. Existing studies already considered the need to implement statistical testing of variables within SDM (Taylor et al., 2009; Ford, 2005), and this application allows for testing and investigating variables' interactions under future uncertainty of water availability conditions and exploring potential impacts, water disputes and crises.

This study focused on the Santa Giustina (S. Giustina) reservoir in the Noce catchment, Autonomous Province of

Trento, Italy, due to its important function for hydropower production and regulating water availability downstream. While water has always been recognized as abundant in the province, temperature increase, loss of glacier mass volume and decreased snow precipitation during winter are among the causes of reduced summer discharge and water availability both in mountain areas and downstream. At the same time, numerous activities have flourished in the Noce catchment such as increasing hydropower plants, agricultural production, urbanization, industrial activities and more intense tourism all demanding large water amounts to satisfy their needs. Tensions for water allocation have recently arisen asking for a fair use of the resource among different actors, also at the same time (e.g. orchard irrigation coinciding with rising tourist water demand). In particular, associations and civil-society groups (e.g. local association for the Noce River safeguard, https://nocecomitato.wordpress.com/, last access: 1 March 2021) were established at the provincial level, showing their concerns about ecological impacts of further exploitation (i.e. hydropower plants). For these reasons, current conditions and future climate change projections leading to critical levels of low and high stored volume and turbined outflows for hydropower production in the S. Giustina dam reservoir were assessed. Indices for frequency, duration and severity of the reservoir's critical states and its reduced water availability are explored and discussed. By doing so, the aim is to develop and demonstrate a stochastic SDM as an effective tool to assess the climate change impact of water scarcity in one of the main reservoirs in the north-east of the Italian Alps supporting its adaptation planning. Such results could inform water operators and local and provincial authorities fostering a discussion on the implementation of climate change adaptation strategies in line with the Water Framework Directive (European Parliament and Council, 2000).

## 2 Case study

The Noce River (Autonomous Province of Trento, Italy) in the south-eastern part of the Alps (Fig. 1) is a tributary of the Adige River, the second longest river in Italy. The Noce River basin is a typical Alpine basin with an overall area of $1367\,\mathrm{km}^2$ and an average discharge of $33.8\,\mathrm{m}^3\,\mathrm{s}^{-1}$ at the basin closure. It is characterized by intensive anthropogenic activities ranging from hydropower plants in the upper part of the catchment relying on glacier melting to intensive apple orchards shaping the landscape of valley bottoms. It also hosts a significant number of tourists with peaks of water demands during winter and summer time for sport activities (i.e. skiing, hiking and kayaking). The hydropower sector is the main water user (77.8 % of the licensed water withdrawals is allocated to small hydropower plants with a nominal capacity $< 3\,\mathrm{MW}$) followed by agriculture (16.4 %), domestic uses (4 %), fish farming (0.9 %), industry (0.4 %),

snowmaking (0.3 %) and others (0.2 %; Provincia Autonoma di Trento, 2018).

Water has always been considered abundant in most regions in the Alps and only recent events of water scarcity in 2015 and 2017 raised wider concerns about water quantity and quality (Stephan et al., 2021; Chiogna et al., 2018; Hanel et al., 2018; Laaha et al., 2017). Within this context, climate change effects at the regional level have already been recognized as acting on the current water balance and triggering multiple impacts on a wide range of economic activities relying on water use (La Jeunesse et al., 2016; Zebisch et al., 2018).

In the Noce River basin, the S. Giustina reservoir provides a large buffer for water resource regulation. The reservoir has a storage capacity of $172\,\mathrm{Mm}^3$ (equal to a maximum net available volume of $152.4\,\mathrm{Mm}^3$), the largest reservoir volume within the Trentino-Alto Adige region. It was built in the 1940s and 1950s for hydropower purposes. Nowadays, the reservoir has a multipurpose function, producing a large amount of energy (i.e. installed power of 108 MW), and regulating water flow for downstream users and providing water for irrigation. Moreover, the local water use plan (Provincia Autonoma di Trento, 2006) established for 2009 onwards a minimum ecological flow threshold ranging from 2.6 to $3.7\,\mathrm{m}^3\,\mathrm{s}^{-1}$ (the only water flow downstream the dam) according to each month of the year to continuously sustain fluvial ecosystems.

## 3 Methods and exploited material

This study focuses on a refinement of SDM applications implementing a stochastic assessment of variables' interactions for validation of uncertainties and trends in the field of risk assessment. Conceptual diagrams of system variables' interactions were elaborated using the Stella software (https://www.iseesystems.com/, last access: 1 March 2021) while statistical correlations, dependencies and tests were analysed in R (Duggan, 2016; R Core Development Team, 2019). This combination contributes to improving SDM analysis accounting for the uncertainty and variability associated with past and future water flow data.

The methodological approach here presented is composed of five sequential phases: (1) the SDM set-up, (2) the analysis of SDM variables' interactions, (3) the SDM model calibration and validation on historical observations, (4) the integration of future projections in the SDM and test of changes of statistical significance to the baseline, and finally (5) the characterization of future low and high stored water critical conditions through a Monte Carlo sampling approach. Each of the stages is described in this section.

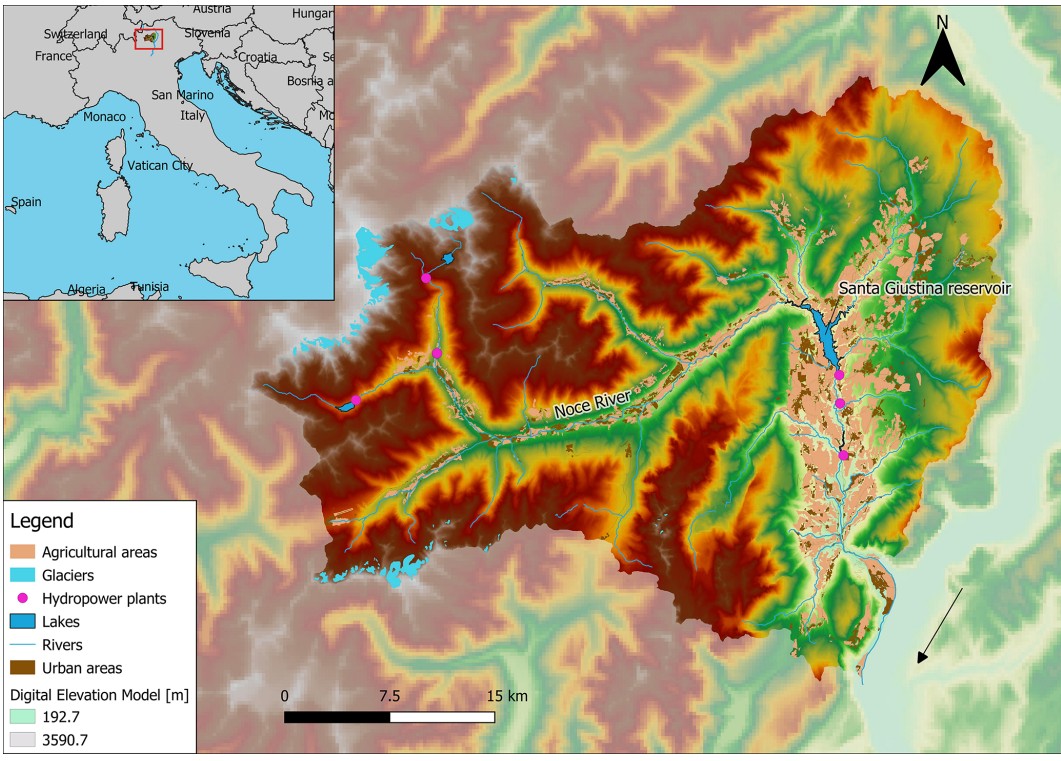

**Figure 1.** Noce River basin and main characteristics. The black arrow specifies the Adige River flow direction.

### 3.1  System dynamics modelling set-up and input data

The SDM was set up by integrating multiple sources of data (e.g. observations, modelled values and climate projections) to replicate past and to simulate future S. Giustina turbined water and stored reservoir volume. The model chain considered for this objective consists of three main modules, namely climate projections in box 1, hydrological model in box 2 and the SDM in box 3 (Fig. 2).

The climate projections provide information stemming from global climate models downscaled to the regional level and bias-corrected through the quantile mapping method (Maraun, 2016; Teutschbein and Seibert, 2012) using the downscaled daily ENSEMBLES daily gridded observational (E-OBS) dataset at 1 km resolution (Cornes et al., 2018) to better simulate climate local conditions. The regional climate model COSMO-CLM (Consortium for Small-scale Modeling Climate Limited-area Modelling) was selected for its spatial resolution of 0.0715° × 0.0715° (≈ 8 km × 8 km), allowing for a local-level climate impact assessment (Rockel and Geyer, 2008). Such model information was developed by the CLM community and provided by the Fondazione Centro Euro-Mediterraneo sui Cambiamenti Climatici (CMCC) as an external component for the application to the Noce catchment (Bucchignani et al., 2016). In the case of climate data from COSMO-CLM, precipitation and temperature baseline data were available from 1971 to 2005; they were also used

to characterize the Noce catchment climatology and compare the baseline with future conditions of precipitation and temperature for the two Representative Concentration Pathways (RCPs).

Temperature and precipitation daily data were used as an input to the physically based model "GEOTRANSF" (external component, Bellin et al., 2016) together with topographical information to replicate streamflow conditions of the Noce River (box 2 in Fig. 2). These two boxes were obtained from simulations developed from the OrientGate project (http://m.orientgateproject.org/, last access: 30 April 2021) and used as an input to focus on the S. Giustina reservoir through the SDM (box 3 in Fig. 2). GEOTRANSF was calibrated and validated on past daily water flow data in the case study area considering a baseline time range from 1981 to 2010 with a performance of the Nash–Sutcliffe efficiency coefficient of 0.88 for calibration and 0.73 for validation at the S. Giustina control section (Bellin et al., 2016). GEOTRANSF provides a description of the water flow within the Alpine Noce River catchment relying on precipitation and temperature data (from box 1), land use and soil type (http://pguap.provincia.tn.it/#, last access: 30 April 2021), streamflow gauge data (https://www.floods.it/, last access: 30 April 2021), and glacier extension (https://webgis.provincia.tn.it/, last access: 30 April 2021). Water withdrawals for anthropogenic uses were modelled within GEOTRANSF at the subcatchment level, accounting for their

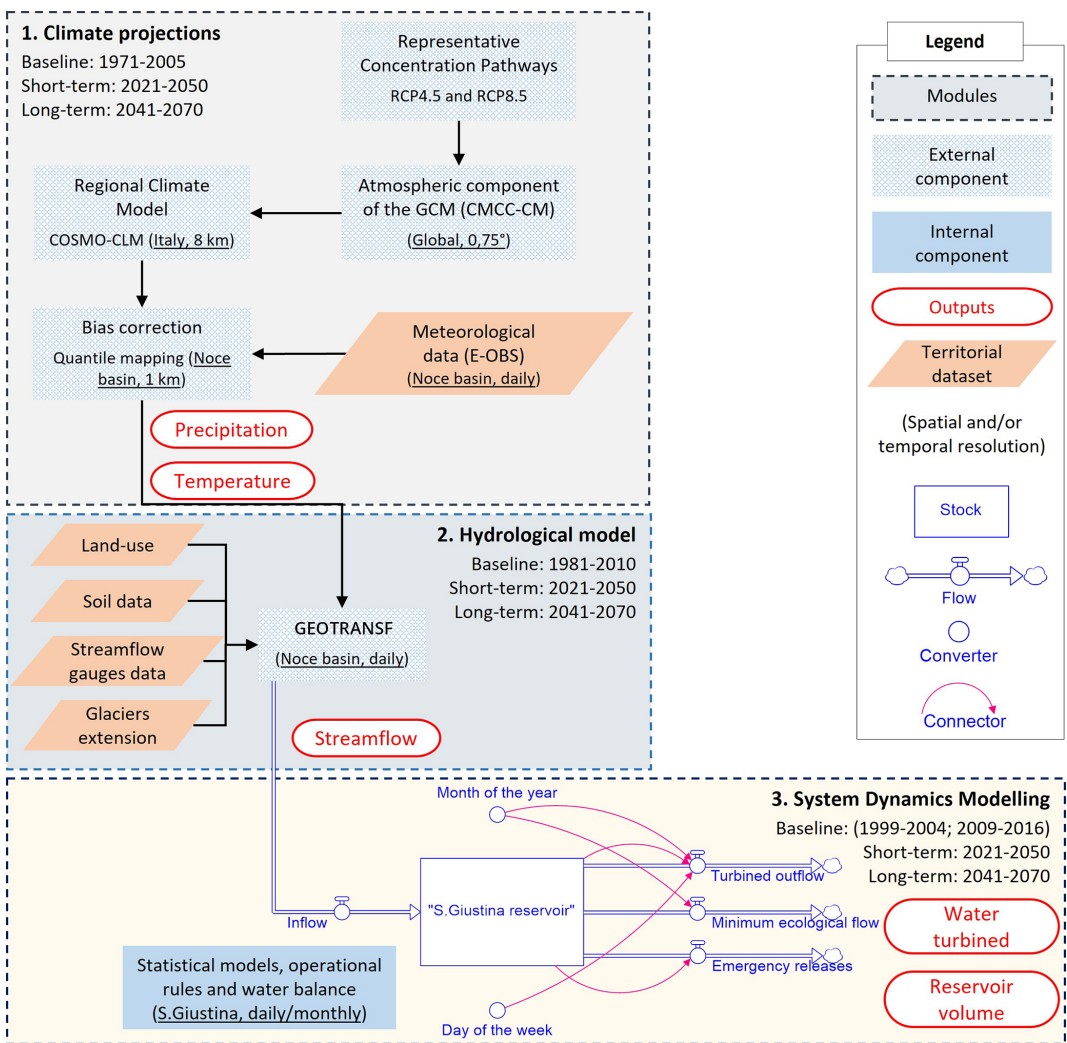

**Figure 2.** Models' chain considered in this study. Box 1 includes the climate modelling used as input for the hydrological model simulation of water streamflow in box 2. Box 3 contains the SDM components of stock (S. Giustina reservoir), flows (inflow and three outflows), converters (month of the year and day of the week) and connectors (red arrows) considered to simulate stored water in the reservoir and the turbined outflow for hydropower production considering a baseline from 1999 to 2016 with a data gap from 2005 to 2008. Sources: adapted from Pham et al. (2019) and Ronco et al. (2017). For an interpretation of the references to colour in this figure legend, the reader is referred to the web version of this article. GCM: global circulation model. CMCC-CM: Centro Euro-Mediterraneo sui Cambiamenti Climatici Climate Model.

licensed number and the physical constraints from the existing water infrastructures. Moreover, GEOTRANSF was applied with COSMO-CLM precipitation and temperature scenarios from 2021 until 2070 over the Noce catchment to assess future conditions of river discharge at the local level for RCP4.5 and RCP8.5 (Bucchignani et al., 2016). While climate was considered the only driver of change in future simulations, anthropogenic water withdrawals were assumed to be at their maximum physical discharge in order to account for possible future increases of water demands from sectors such as domestic and agricultural needs, sectors recognized to lead to possible increases in demand (Bellin et al., 2016; La Jeunesse et al., 2016).

The stochastic SDM relied on the inflow values from GEOTRANSF applications as one input variable. Other input variables were initially considered and tested as other variables data in the SDM to replicate turbined water and stored volume (excluded to the modelling; further information reported in the Supplement). The baseline simulation period was constrained by the reservoir volume data availability over 14 years with a range 1999–2004 and 2009–2016 and a data gap from 2005 to 2008. For this reason, the SDM was forced with past observations of inflow to S. Giustina (with data available for the period 1981–2016) and with GEOTRANSF values for future simulations due to the limited temporal overlap of past GEOTRANSF values (over 1981–

**Table 1.** Selected variables within the SDM (box 3 in Fig. 2) for the S. Giustina reservoir.

| Data type | Variable name | Time range | Source |
|---|---|---|---|
| Inflows to S. Giustina [$m^3\,s^{-1}$] | Inflows | 1981–2016 | |
| S. Giustina outflows for hydropower use [$m^3\,s^{-1}$] | Outflow | 1981–2016 | |
| S. Giustina volume [$Mm^3$] | Volume | 1999–2004 2009–2016 | Autonomous Province of Trento – Agency for Water Resources and Energy |
| Minimum ecological flow [$m^3\,s^{-1}$] | MEF | 1999–2016 | |
| Emergency releases [$m^3\,s^{-1}$] | Releases | 1981–2016 | |

2010) with observations of the S. Giustina stored volume (1999–2004 and 2009–2016, Table 1). The SDM was run at a daily time resolution to replicate the different outflow's regulations (e.g. minimum ecological flow and emergency outflows). Finally, simulations were aggregated to monthly values being a suitable time resolution for supporting reservoir volume management over long periods considering climate change effects (Solander et al., 2016).

### 3.2 Variables' interaction analysis

This analysis aims to statistically describe the existence and type of interactions among the system variables. For the simulations of turbined outflows, different statistical models' and variables' interactions were tested and recursively implemented to simulate the stored water in S. Giustina by applying the reservoir water balance equation.

#### 3.2.1 Hydropower outflows

The simulation of turbined outflows from the S. Giustina reservoir for hydropower production considered regressions of different input variables (e.g. inflow, hydroelectric energy market price, temperature, precipitation and water outflows from an upstream dam reservoir) and statistical models, including linear regressions and more flexible generalized additive models.

After an initial exploratory data analysis, the variables' hydroelectric energy market price, temperature, precipitation and water outflows from an upstream dam reservoir were rejected based on criteria of (i) data availability to the maximum target time period, (ii) correlations among the explanatory and response variables, and (iii) the selection of the most parsimonious model. Further information on input variables, their tested combinations for model selection and their link to the open code is reported in the Supplement.

The best regressions for each model type are reported in Table 2. The "lme4" package in R (Bates et al., 2015) was applied for linear mixed-effects models, while the "mgcv" package in R (Wood, 2017; Wood and Scheipl, 2020) was used for the generalized additive models. Model 2 in Table 2 was selected as the best regression for its performance of 0.75 for the adjusted $R^2$ and 16.57 $Mm^3$ for the RMSE at a monthly scale (daily scale of 0.42 for the adjusted $R^2$ and 0.95 $Mm^3$ for the RMSE). In particular, the linear mixed-effects model showed its ability to account for weekday and monthly variations as an important variable to capture weekly and seasonal reservoir management while showing lower proneness to overfit calibration data compared to flexible non-linear models. The model simulated water diverted to the turbines ($Q_{out}$) as a function of water flowing into the reservoir ($Q_{in}$), the volume state in the previous day ($V(t-1)$, through the "lag" operator to shift the time series back by one time step), and the day of the week and the month of the year. As a fixed effect the water flowing into the reservoir and the volume in the previous day were here considered to account for the linear relation with the turbined water. As a random effect to capture differences among different groups, day of the week and month of the year were selected to account for the recurrent water volume variations occurring according to the weekday group and month group (distinguished by vertical bars in linear mixed-effects models). By doing so, it was possible to describe the reservoir water volume by combining the GEOTRANSF model outputs with statistical analysis aiming to explore the reservoir volume vulnerability to future changing conditions.

#### 3.2.2 Reservoir volume

Simulated outflow values were used at each time step to replicate the volume stored through the water balance equation

**Table 2.** Best of each model type and their performance indicators ($R^2$ and the root mean square error, RMSE) at monthly resolution and at daily resolution in brackets. The best selected model is reported in bold. The syntax follows that of the R packages "lme4" (for the linear mixed-effects model) and "mgcv" (for the generalized additive models) with the "bs" term set to "re" to refer to the random effect in the generalized additive mixed model; "s" is the function used in definition of smooth terms within the gam model formulae. A full list of tested models and their features is reported in the Supplement.

| Model types | No. | R syntax | Monthly resolution (daily) | |
| --- | --- | --- | --- | --- |
| | | | Adjusted $R^2$ | RMSE ($\times 10^6$) |
| Multi-linear model | 1 | lm(Outflow∼ Inflows + lag(Volume)) | 0.63 (0.35) | 19.86 (1.03) |
| **Linear mixed-effects model** | **2** | **lmer(Outflow∼ Inflows + lag(Volume) + (1\|weekday) + (1\|month))** | **0.75 (0.42)** | **16.57 (0.95)** |
| Generalized additive model | 3 | gam(Outflow∼ s(Inflows) + s(lag(Volume))) | 0.61 (0.50) | 20.31 (1.07) |
| Generalized additive mixed model | 4 | gam(Outflow∼ s(Inflows) + s(lag(Volume)) + s(weekday, bs = "re") + (month, bs = "re")) | 0.65 (0.54) | 19.27 (1.02) |

Eq. (1):

$$\frac{dV(t)}{dt} = Q_{in}(t) - Q_{out}(t) - Q_{mef}(t) - Q_{rels}(V), \qquad (1)$$

where $Q_{in}$ is the water flowing into the reservoir, $Q_{out}$ is the water diverted to the turbines, $Q_{mef}$ is the minimum ecological flow released and $Q_{rels}$ is the water outflow for emergency flood releases. In particular, $Q_{mef}$ was set to $2.1\,\mathrm{m^3\,s^{-1}}$ in the case of simulations before 2004, while it is equal to $2.6\,\mathrm{m^3\,s^{-1}}$ for January–March and December; $3.68\,\mathrm{m^3\,s^{-1}}$ for April–July, October and November; and $3.2\,\mathrm{m^3\,s^{-1}}$ for August and September from 2009–2016 (http://pguap.provincia.tn.it/#, last access: 30 April 2021). Moreover, the same values of minimum ecological flow were set in future simulations assuming no changes in minimum discharges for the ecological flow in the future. A simplified operational rule was implemented for emergency releases from flood gates considering the maximum daily emergency discharge value of $168.5\,\mathrm{m^3\,s^{-1}}$ in the case of daily stored volume greater than or equal to $159.3\,\mathrm{Mm^3}$.

## 3.3 Model calibration and validation

The statistical model for turbined water was calibrated and validated over 5061 d of available data for the baseline period, representing a total of 14 years from 1999 to 2004 and from 2009 to 2016. A forward time window approach was applied as a cross-validation technique to better estimate model fitting (i.e. based on training data) and predictive performance (i.e. based on temporally independent test data) using the root mean square error (RMSE). The applied methodology is based on multiple separations of training and testing datasets. Within the first repetition, the predefined model set-ups (i.e. turbined outflows models) are calibrated using a subset of the original data that relates to the first 3650 d of available data. The derived relationships are then tested using both training data (i.e. fitting performance) and the dataset that relates to the remaining (not yet) considered days (i.e. predictive performance). The following 1410 repetitions are based on the same procedure but on increasingly larger training datasets (i.e. consecutively adding 1 d within the forward

time window approach). The mean value for RMSE was calculated considering all 1411 repetitions over the increasingly larger dataset, providing a more robust estimation of model performance and its variability using multiple temporally independent subsets of the original data (Hastie et al., 2009; Kohavi, 1995; Tashman, 2000; Varma and Simon, 2006). This methodology allows for overcoming some limitations of common one-fold non-temporal validation methods (splitting of training and test data randomly; e.g. hold-out validation) associated with data temporal dependencies (i.e. autocorrelation) and an arbitrary choice of training and validation subsets. A major advantage of such multi-fold partitioning strategies is the possibility to exploit all the available data for the generation of the final prediction model.

## 3.4 Future projections and statistical testing

Future water inflow to the reservoir (coming from the GEO-TRANSF application) was used to simulate future turbined outflow and volumes stored in the S. Giustina reservoir. GEOTRANSF simulations considered unchanged maximum water withdrawals in the Noce catchment in the future, and although possible variations in the future may affect river water flows, this set-up provided a conservative assumption accounting for future water demand increases, such as from domestic and agricultural needs, in future scenarios. Moreover, integrated downscaled COSMO-CLM climate scenarios were considered (Bellin et al., 2016; Bucchignani et al., 2016). Such climate projections have been demonstrated to represent climate forcing variables (i.e. precipitation and temperature) over Alpine regions well (Montesarchio et al., 2013).

The RCP4.5 and RCP8.5 scenarios were selected according to the IPCC AR5 (Intergovernmental Panel on Climate Change Fifth Assessment Report; IPCC, 2014). Simulations stretched over two 30-year time horizons to represent short-term (2021–2050) and long-term (2041–2070) future climate conditions affecting the Noce River flow and the S. Giustina reservoir management.

Differences between the future and baseline simulated turbined water and volume stored were statistically tested through the Wilcoxon rank sum test, a non-parametric test selected since it allows for dealing with non-normally distributed, unpaired groups of data (Hollander and Wolfe, 1973). It was implemented to test whether future predicted values of stored volume were significantly different than the baseline and to reject the null hypothesis (i.e. same tendencies among the tested groups) in the case of a $p$ value $\leq 0.05$.

## 3.5    Characterization of future critical conditions

This study explored and analysed critical conditions of low and high future stored volume considering the 10th and 20th and 80th and 90th percentile thresholds calculated from water stored from the baseline (1999–2004 and 2009–2016). Such thresholds provided a symmetrical reference of extreme conditions and were already identified in previous studies as significant levels to assess critical states in hydrological studies (Yilmaz et al., 2008). Considering these thresholds, a set of different metrics were calculated to characterize the frequency, duration and severity of future critical volume conditions (Table 3) based on existing metrics used to describe extreme events (Vogt et al., 2018). In particular, frequency was considered in relative terms as the ratio of the overall number of months below or above the selected threshold used to define the level of volume critical conditions over 14 years, hence describing yearly conditions of extreme events. Maximum duration was described considering the maximum number of consecutive months below or above the selected thresholds. The severity metric describes the accumulated deficit/surplus of simulated volume values with respect to the total volume stored over a 14-year window (relative severity). This last metric provides information on the fraction of the total stored volume in deficit or surplus as the average annual severity.

Moreover, a Monte Carlo approach was implemented to account for the uncertainty related to the simulations in future conditions. The water balance equation considered the lower and upper turbined outflow from the prediction bands to generate a range of simulated volume values. A set of 1000 replications of 30-year length per each future climate scenario was generated by randomly sampling from the simulated volume values and their prediction bands. In addition, for each replication a subset was iteratively extracted considering a time window of 14 years moving progressively at a monthly time step along the simulated 30 years of future volume. Extreme condition metrics of relative frequency, maximum duration and relative severity were calculated on the subset at each iteration. This procedure allowed for comparing the calculated metrics of each subset replication with the 14 years of available data for the baseline volume (1999–2004 and 2009–2016). By doing so, it was possible to account for the uncertainty related to the modelling and pro-

vided a wider range of low and high future volume values for a more robust characterization of their conditions.

## 4    Results

### 4.1    Baseline period

The linear mixed-effects model was used to replicate observations of turbined water outflows from the S. Giustina reservoir (Fig. 3). The model was run at a daily time step, and values were aggregated and reported at a monthly resolution. The model gave an $R^2$ of 0.75 and a mean RMSE of 16.57 Mm$^3$. Figure 3 shows the modelled and real values, with the $y$ scale ranging from 0 % (i.e. no turbined water) to 100 % (i.e. maximum turbined water of 176 Mm$^3$ per month for 31 d of full turbine operations). Coloured bands outline areas above or below the percentile values defining critical thresholds that were considered throughout the analysis.

The modelled turbined water was then used in the iterative implementation of the water balance equation together with the operational rules for the minimum ecological flow and the emergency releases. Figure 4 shows the modelled and real volume ranging from 0 % to 100 % of stored volume (equal to 159.3 Mm$^3$, maximum volume allowed for flood prevention), where the simulation of the stored volume resulted in an $R^2$ of 0.60 and a mean RMSE of 19.74 Mm$^3$ per month. In Figs. 3 and 4 the general behaviours of real turbined outflows and stored water were replicated by the regression models. Some specific very high and low conditions were not completely represented or missed due to abrupt changes in the reservoir management, such as the 2001 summer peak of turbined outflows in Fig. 3, due to persistent inflows to S. Giustina forcing dam managers to turbine at the maximum outflow for 23 d consecutively, as well as the low values in spring 2003 in Fig. 4, when the S. Giustina reservoir was emptied due to construction works on a penstock.

### 4.2    Future projections and statistical testing

Future GEOTRANSF model results forced by the COSMO-CLM climate projections depict a situation of general decreases in precipitation and water inflowing to the reservoir (Table 4 and Fig. 5). In particular, changes in RCP4.5 are rather similar for the short and long terms, whereas changes are larger in the long term for RCP8.5 compared to the short term. These results are consistent with the precipitation trends from COSMO-CLM projections with RCP8.5 showing higher values of total precipitation in the short term but a higher decrease in the long term compared to RCP4.5 (Bucchignani et al., 2016). In RCP8.5, inflow, outflow and volume reductions are lower for the short-term future compared to the baseline ($-7.8$ %, $-11.5$ % and $-10.2$ %) and are associated with the only case of precipitation increase ($+1.4$ %), pointing to the increase of evapotranspiration due to the relatively larger increase in temperature ($+29.4$ %). In

**Table 3.** Metrics implemented to characterize extreme events of low and high volume stored in the S. Giustina reservoir for RCP4.5 and RCP8.5 future projections, adapted from Vogt et al. (2018).

| No. | Metrics | Unit | Description |
|---|---|---|---|
| 1 | Relative frequency | Months/year | Number of months below or above the selected thresholds used to define the level of volume critical conditions over a 14-year period |
| 2 | Maximum duration | Months | Maximum number of consecutive months below or above the selected thresholds used to define the level of volume critical conditions |
| 3 | Relative severity | % | Sum of the differences, in absolute values, between simulated volume values and the selected thresholds over the total stored volume and 14-year period; $S_i = \frac{\sum |V_i| < \text{Threshold}}{\sum V_i \cdot 14}$ |

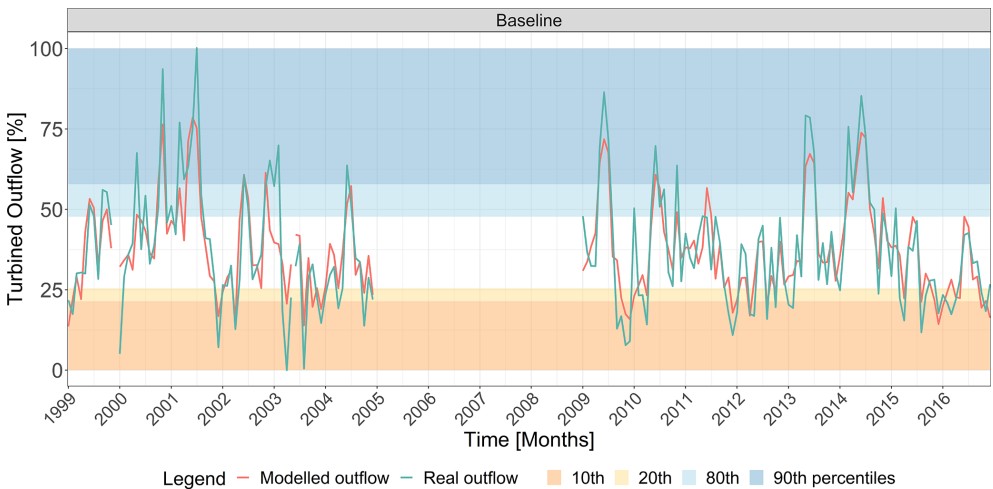

**Figure 3.** S. Giustina water diverted to the turbines from 1999 to 2016. Modelled (red line) and real (green line) values. Adjusted $R^2 = 0.75$, mean RMSE $= 16.57\,\text{Mm}^3$. Coloured bands outline areas of values lower than the 10th and 20th and greater than the 80th and 90th percentiles. For an interpretation of the references to colour in this figure legend, the reader is referred to the web version of this article.

the long term, results show the greatest increase of temperature ($+58.8\,\%$) and reduction of precipitation ($-4.3\,\%$) as well as of inflow, outflow and volume ($-21.3\,\%$, $-26.2\,\%$ and $-20.8\,\%$).

A summary overview of future conditions for inflow, turbined water and stored volume is reported in Fig. 5. For all variables, four coloured bands representing areas lower than the 10th and 20th and greater than the 80th and 90th percentile are reported as a reference, which allows for comparing the 30-year time slices. Values show a generalized decrease into the 20th and 10th percentile thresholds of the long-term RCP8.5 scenario. This trend is clearly visible for future water inflow, while values larger than boxplots whiskers can be identified in all inflow scenarios and hence point to single future conditions greater than baseline maximum recorded values (Fig. 5a). Future turbined outflows show boxplot values with the lowest interquartile range expected to reach 10th percentile conditions. Consistently with

the results from Table 4, volume values show significant reductions already for RCP4.5 with interquartile levels below with the 20th percentile calculated from the baseline (i.e. 1999–2004 and 2009–2016). These results were further investigated through the application of the Wilcoxon rank sum test (Table 5), which provides more quantitative insights into the statistical significance of changes considering the whole 30-year period of data for the baseline and the four scenarios (Fig. 5) as well as considering monthly averaged values (Figs. 6 and 7). While $p$ values considering the whole time series are below the 0.05 threshold of significance, monthly averages for the short-term RCP8.5 scenario (2021–2050) provided non-significant results both for stored volume and turbined outflow.

Results of future turbined outflows are reported in Fig. 6 with values averaged for each month over the 30-year simulation and compared to the baseline (i.e. percentage change). Negative reductions of outflows for all scenarios are reported

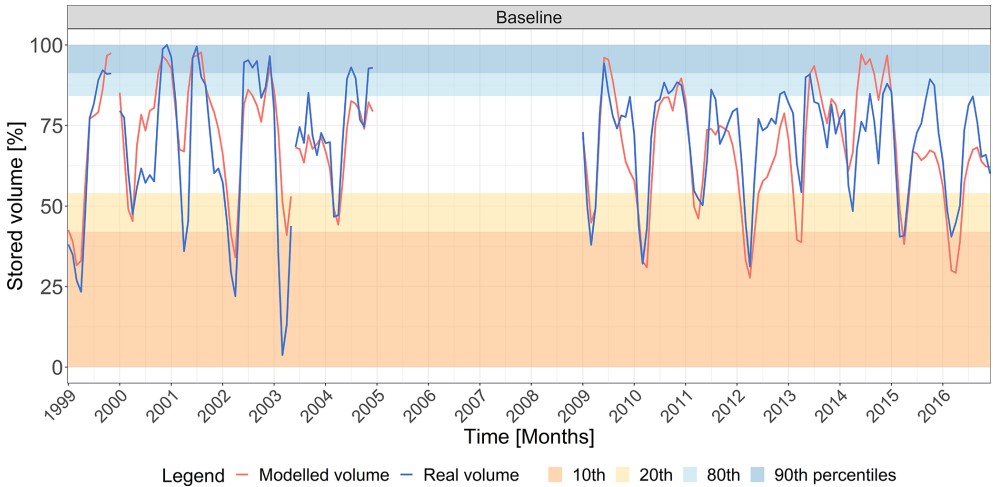

**Figure 4.** S. Giustina stored volume values from 1999 to 2016. Modelled (red line) and real (blue line) values. Adjusted $R^2 = 0.60$, RMSE $= 19.74$ Mm$^3$. Coloured bands outline areas of values lower than the 10th and 20th and greater than the 80th and 90th percentiles. For an interpretation of the references to colour in this figure legend, the reader is referred to the web version of this article.

**Table 4.** Average values of temperature and precipitation (COSMO-CLM projections), water inflow to the S. Giustina reservoir, turbined outflow, stored volume (simulations) and their percentage differences compared to baseline values.

| Variable | Baseline | RCP4.5 | | | | RCP8.5 | | | |
|---|---|---|---|---|---|---|---|---|---|
| | * | 2021–2050 | | 2041–2070 | | 2021–2050 | | 2041–2070 | |
| | Value | Value | Δ [%] | Value | Δ [%] | Value | Δ [%] | Value | Δ [%] |
| Temperature [°C] | 5.1 | 6.5 | +27.5 | 7.5 | +47.1 | 6.6 | +29.4 | 8.1 | +58.8 |
| Precipitation [mm yr$^{-1}$] | 1495.1 | 1433.6 | −4.1 | 1391.5 | −6.9 | 1516.3 | +1.4 | 1430.7 | −4.3 |
| Inflow [Mm$^3$ per month] | 71.5 | 57.4 | −19.7 | 58.3 | −18.5 | 65.9 | −7.8 | 56.3 | −21.3 |
| Turbined outflow [Mm$^3$ per month] | 64.1 | 48.2 | −24.8 | 48.7 | −24 | 56.7 | −11.5 | 47.3 | −26.2 |
| Stored volume [Mm$^3$] | 109.6 | 87.8 | −19.9 | 88.2 | −19.5 | 98.4 | −10.2 | 86.8 | −20.8 |

* Baseline period for climate data goes from 1971 to 2005, while for water inflow and volume stored it spans over 1999–2004 and 2009–2016 (14 years).

for spring and summer, starting in April and going until September with differences up to −56.3 % in August for the long-term RCP4.5 scenario. All climate scenarios agree on a water flow reduction during November reaching a minimum of −33.5 % of turbined outflow for the long-term RCP8.5 scenario. In all other months, scenarios depict varying conditions of water flow. In particular, the short-term RCP4.5 scenario depicts conditions of negative differences for every month of the year. An increased number of positive differences are predicted for the long-term RCP4.5 scenario during January (+5 %) and December (+5.3 %). The short-term RCP8.5 scenario shows larger positive differences during January (+10.3 %), February (+1.8 %), March (+2.9 %), October (+0.8 %) and December (+6.5 %). The long-term RCP8.5 scenario projects a negative trend from April until the end of the year, reaching persistent negative conditions in summer down to −55.5 % in August, overlapping with the summer electricity peak loads and calling for particular attention (Terna, 2019). Nevertheless, small but positive values are expected for January (+0.7 %), February (+1.4 %)

and March (+3.7 %), when the winter electricity peak load usually occurs (Terna, 2019).

Comparative results with the baseline for the monthly average over the 30-year simulation are reported for the stored volume in Fig. 7. All climate scenarios agree on the general volume decrease from May until the end of the year. The short- and long-term RCP4.5 scenario depicts conditions of minimum peaks in May (−43.8 % and −40.6 %) and June (−44.1 % and −41.9 %), while the long-term RCP8.5 scenario shows fewer negative minimum values, although persistent negative values in November and December lower than the other scenarios (−26.8 % and −19 %). Scenarios agree on the positive variation during February and March with a maximum increase of +32.5 % for the short-term RCP8.5 case. However, scenarios disagree in terms of stored volume for January, with RCP4.5 scenarios representing a positive variation both in the short- and long-term cases (+1.4 % and +2.2 %), while the short-term RCP8.5 case depicts a positive variation (+7.3 %) but a negative one for the long-term case (−7.1 %). Conditions in April are reversed

**Table 5.** Summary of the Wilcoxon rank sum test application to stored volume and turbined outflow for the four scenarios compared to the baseline (1999–2004 and 2009–2016); $p$ values are reported for the test considering the whole time series of future and baseline values and on paired monthly averages.

| Compared scenarios | Variable | On the whole time series $p$ value | On monthly averages $p$ value |
|---|---|---|---|
| RCP4.5 2021–2050 vs. baseline | Stored volume | $2.203 \times 10^{-15}$ **** | 0.016* |
| | Turbined outflow | $3.737 \times 10^{-12}$ **** | 4.883e-4*** |
| RCP4.5 2041–2070 vs. baseline | Stored volume | $2.275 \times 10^{-16}$ **** | 0.034* |
| | Turbined outflow | $8.676 \times 10^{-12}$ **** | 0.009** |
| RCP8.5 2021–2050 vs. baseline | Stored volume | $1.120 \times 10^{-6}$ **** | 0.092 |
| | Turbined outflow | 0.003** | 0.095 |
| RCP8.5 2041–2070 vs. baseline | Stored volume | $5.006 \times 10^{-18}$ **** | 0.012* |
| | Turbined outflow | $1.381 \times 10^{-13}$ **** | 0.007** |

Symbols *, **, *** and **** refer to significant $p$ values $\leq 0.05$, $\leq 0.01$, $\leq 0.001$ and $\leq 0.0001$.

with the short-term RCP4.5 scenario depicting a decrease ($-8.3\%$) and RCP8.5 increases for short- and long-term cases ($+5.5\%$ and $+3.5\%$).

Months of positive variation and scenario disagreement provide important information on the timing of potential reservoir management adaptation, while the small volume increases are insufficient to counterbalance persistent volume reductions. The short-term RCP8.5 scenario shows the most favourable conditions of water volume, depicting positive differences in January ($+7.3\%$), February ($+22.5\%$), March ($+32.5\%$) and April ($+5.5\%$).

## 4.3 Characterization of future critical conditions

Critical conditions of stored reservoir water volumes (both high and low) were explored to further understand how climate change may impact long-term reservoir vulnerability. A set of different metrics were calculated to characterize frequency, duration and severity of future critical volume conditions considering values lower than the 10th and 20th and higher than the 80th and 90th percentiles of stored volume (Figs. 8 and 9). The metrics were calculated from 1000 replications per scenario randomly sampled from the simulated future volume values and their prediction bands. Reductions were shown to have similar trends across the metrics between the 10th and 20th as well as between the 80th and 90th percentiles, with the four future scenarios having statistically significant differences compared to the baseline.

Boxplots for low-volume conditions (Fig. 8) show increasing average values for all metrics and scenarios compared to the baseline: conditions of low volume are expected to become more frequent, having a longer maximum duration and larger severity. In particular, for both values lower

than the 10th and 20th percentile, the short-term RCP4.5 scenario shows the highest average increase. Relative frequency and relative severity are expected to have a higher increase for average values below the 10th percentile ($+157\%$ for frequency, from 1.64 to 4.21 months $yr^{-1}$, and $+250\%$ for severity, from 0.2% to 0.7%) compared to values below the 20th percentile ($+105\%$ for frequency, from 2.86% to 5.86%, and $+300\%$ for severity, from 0.3% to 1.2%), while maximum duration is expected to have a higher increase for values below the 20th percentile ($+100\%$, from 5 to 10 months) compared to values below the 10th percentile ($+75\%$, from 4 to 7 months). These results point to events of low-volume conditions below the 10th percentile becoming more frequent and with a higher severity, while low-volume events below the 20th percentile are expected to last for a longer time in the case of the most extreme events. RCP8.5 in the short term depicts fewer negative conditions for both thresholds in line with results in Fig. 6: for values below the 10th percentile, $+40\%$ in relative frequency (from 1.64 to 2.29 months $yr^{-1}$), $+0\%$ in maximum duration (4 months) and $+50\%$ in relative severity (from 2% to 3%) and for values below the 20th percentile, $+38\%$ in relative frequency (from 2.86 to 3.93 months $yr^{-1}$), $+20\%$ in maximum duration (from 5 to 6 months) and $+67\%$ in relative severity (from 0.3% to 0.5%).

For values lower than the 20th percentile, a larger number of values outside the boxplot whiskers are depicted for all scenarios and especially towards higher number of months for relative frequency as well as maximum duration and relative severity, hence pointing to potential single conditions of lower volume stored in the future with respect to the represented median. These results point to the possibility of the most extreme water scarcity conditions lasting

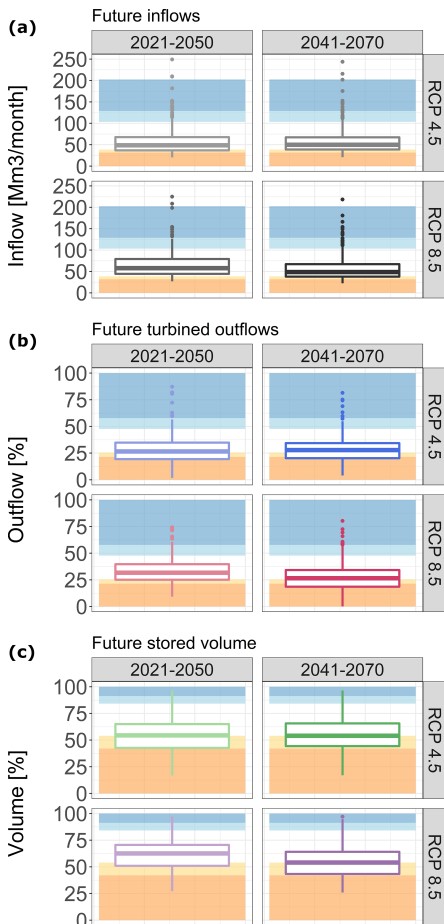

**Figure 5.** Thirty-year slice boxplots for future projections of **(a)** simulated water inflow to the S. Giustina reservoir, **(b)** simulated turbined water and **(c)** simulated future water volume stored in the S. Giustina reservoir. Coloured bands outline areas of baseline values lower than the 10th and 20th and greater than the 80th and 90th percentiles using the same colours as in Fig. 4. For an interpretation of the references to colour in this figure legend, the reader is referred to the web version of this article.

longer than 1 hydrological year and pointing to chronic consequences of low stored volume, especially considering the short-term RCP4.5 scenario (maximum values of 19 months and 31 months for values below the 10th and 20th percentile) and the long-term RCP8.5 scenario (maximum values of 15 and 30 months for values below the 10th and 20th percentile), where the highest outlier values are expected.

Boxplots for high-volume conditions (Fig. 9) also show a generalized decrease in all metrics of high stored volume for all scenarios and for both thresholds. The long-term RCP4.5 and RCP8.5 scenarios depict conditions of higher percentage reductions for relative frequency (−75 % for both scenarios) and maximum duration (−60 % for both scenarios) above the 90th percentile compared to the 80th percentile (−68 % in relative frequency and −50 % in maximum duration for both scenarios). On the contrary, relative severity

reductions are expected to be higher for RCP8.5 for values above the 80th percentile (−97 %) compared to the 90th percentile (−80 %). Above the 90th percentile, values are expected to be less frequent than baseline conditions, while they show smaller severity reductions compared to values above the 80th percentile. The short-term RCP8.5 scenario predicts a smaller decrease in all metrics in comparison with the other scenarios with −48 % of relative frequency (from 3.14 to 1.64 months yr$^{-1}$), −50 % of maximum duration (from 6 to 3 months) and −75 % in relative severity (from 0.4 % to 0.1 %) for values above the 80th percentile compared to the baseline, as well as −56 % of relative frequency (from 2.29 to 1 months yr$^{-1}$), −60 % of maximum duration (from 5 to 2 months) and −68 % of relative severity (from 0.3 % to 0.08 %) for values above the 90th percentile compared to the baseline. A summary table with median, maximum, minimum and standard deviation values for low- and high-volume conditions is provided in the Supplement.

## 5  Discussion

The analysis considered the water flowing into the S. Giustina reservoir and modelled using the GEOTRANSF hydrological model as a key variable influencing the turbined water and hence the stored volume. A stochastic approach considered the simulated turbined outflows and its prediction bands as the main source of uncertainty to explore a wide range of possible outcomes in terms of turbined outflow and stored volume values.

Results show how the amount of water flowing into the reservoir is deeply affecting both turbined outflows and hence the stored water, which are expected to reduce in the future even under the short-term RCP4.5 scenario (−25.9 %). In the case of long-term scenarios, high reductions are also expected (−24 % for RCP4.5 and −26.3 % for RCP8.5).

Moreover, results on monthly averages provide useful information on the timing of possible consequences coming from reservoir operations and climate change effects. Considering those months of positive variation of volume (i.e. January, February, March and April, depending on the considered scenario) and turbined water outflows (i.e. December, January, February and March, depending on the considered scenario) provides insights into the need to plan adaptation and operational strategies to improve the management of the S. Giustina reservoir. Months of positive water volume changes need to be considered periods of preparation to worsening conditions. Strategies of earlier water accumulation should be considered to face the persistent reductions throughout the last part of spring, summer, autumn and part of winter. Such a strategy could prevent downstream conditions of water shortages while also preparing for reductions in turbined water for hydropower use, especially during summer and winter months of high electricity peak loads. These results are in line with other findings in the Alps showing

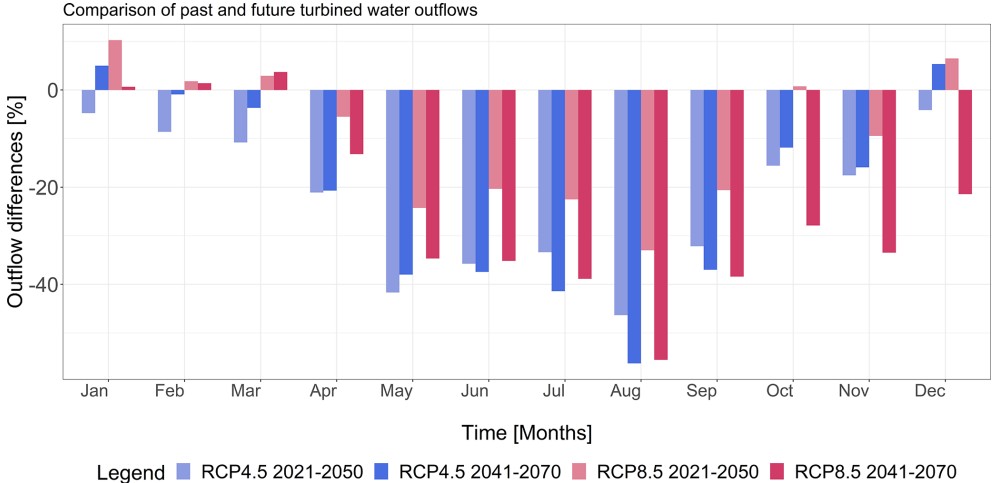

**Figure 6.** Percentage change of turbined water outflows [%] comparing the four climate scenarios to the baseline at a monthly level. For an interpretation of the references to colour in this figure legend, the reader is referred to the web version of this article.

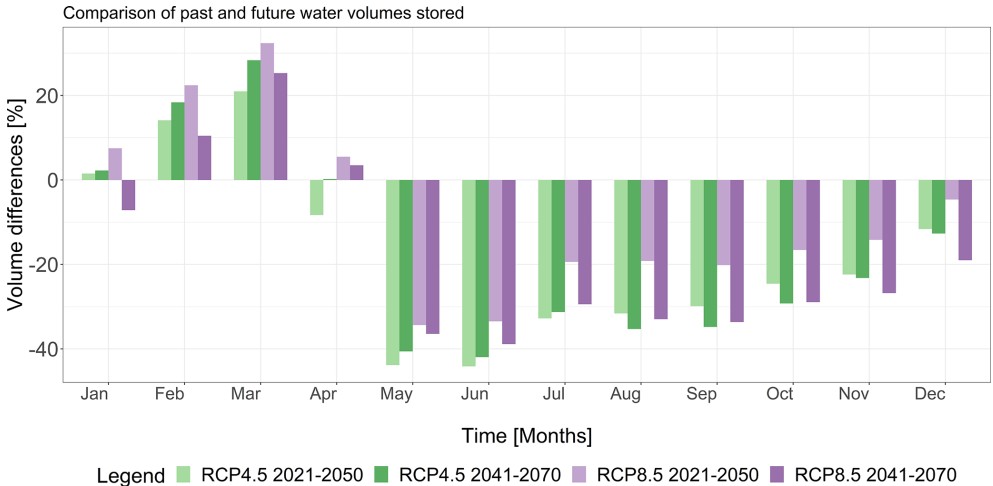

**Figure 7.** Percentage change of volume [%] comparing the four climate scenarios to the baseline at a monthly level. For an interpretation of the references to colour in this figure legend, the reader is referred to the web version of this article.

the need for earlier reservoir water accumulation during winter to prevent downstream conditions of water shortages during summer (Brunner et al., 2019; Hendrickx and Sauquet, 2013). Although positive percentage variations are expected to be lower and for fewer months than negative cases, earlier water accumulation strategies for potentially reducing water scarcity conditions need to acknowledge and avoid an exacerbation of flood events. Additional storage for flood prevention needs to be ensured and managed together with the Civil Protection Department to prevent potential downstream floods.

Negative stored volume variations are supported by the generalized trends of future water scarcity conditions characterized by an increase in frequency, maximum duration and severity of low stored volumes for values lower than the 10th and 20th percentiles and for both RCP4.5 and RCP8.5. These

results point at higher percentage increases in frequency and severity for values below the 10th percentile, while volume values below the 20th percentile are expected to last longer in the case of most extreme events. At the same time, high-volume conditions decrease in terms of frequency, duration and severity with higher reductions for volume above the 90th percentile compared to the percentage decrease for events above the 80th percentile. Only in the case of relative severity are reductions expected to be higher for values above the 80th percentile compared to those above the 90th percentile for the long-term RCP4.5 and RCP8.5 scenarios. Above the 90th percentile, values are expected to be less frequent than in baseline conditions while showing smaller severity reductions compared to values above the 80th percentile. Within this context, the calculation of a set of metrics in terms of low- and high-volume conditions through a

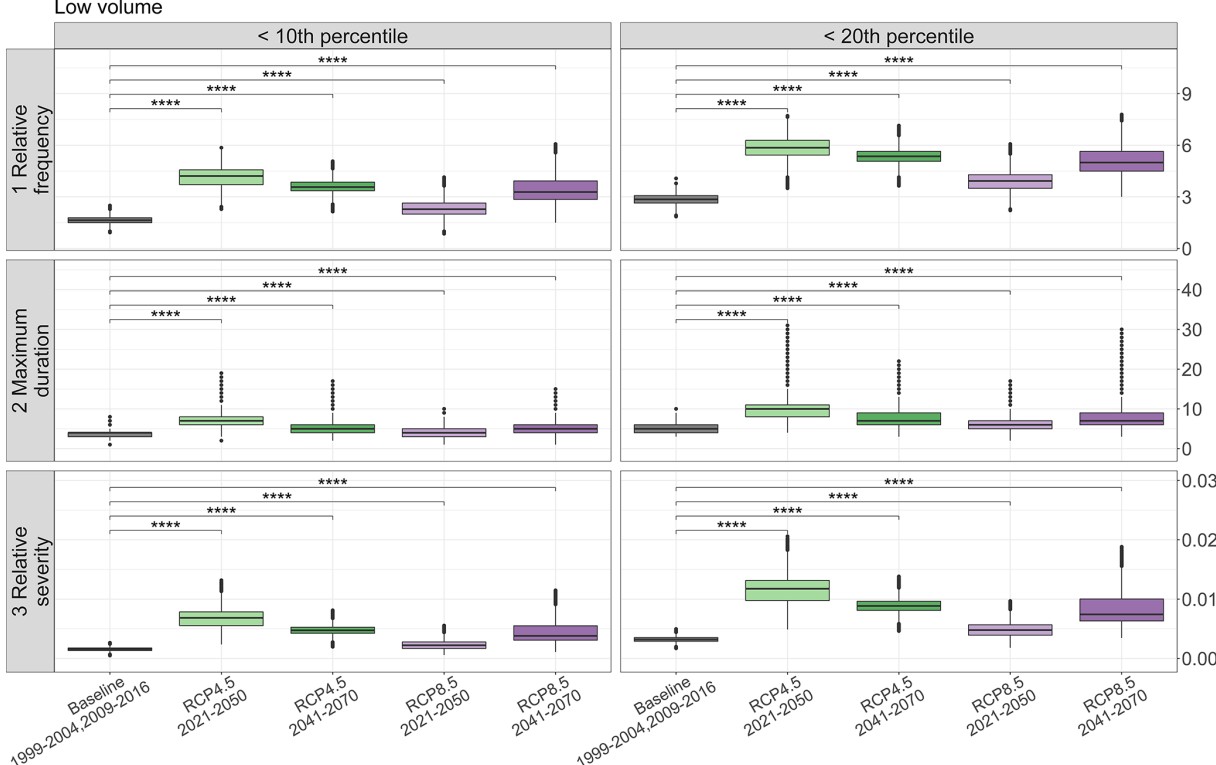

**Figure 8.** Boxplots of metrics of relative frequency, maximum duration and relative severity calculated from volume values lower than the 10th and 20th percentiles. The simulated volume values used for the metric calculation resulted from the Monte Carlo approach by randomly sampling from the volume prediction bands. The symbol **** refers to a $p$ value $\leq 0.0001$. For an interpretation of the references to colour in this figure legend, the reader is referred to the web version of this article.

Monte Carlo approach considering a moving window of data (Figs. 8 and 9) allowed for generating a set of possible future volume time series derived from the simulation prediction bands. By doing so, the analysis prevented any potential bias associated with limited data and provided statistically tested information on increases of low-volume and reductions of high-volume values shown as boxplots for S. Giustina over four 30-year climate scenarios.

In general, the results suggest exacerbated risks to reservoir operation due to persistent stored volume and turbined outflow reductions in late spring and summer, autumn, and early winter that can potentially lead to chronic consequences lasting more than 1 hydrological year and hence threatening water supply security, hydropower production and ecosystem services in the valley.

## 5.1  Limitations of the study

The applied SDM mainly considers outputs from the GEO-TRANSF applications integrating the COSMO-CLM climate projections. However, several assumptions and limitation in this study are noted.

Accounting for the GEOTRANSF application means not only relying on a very accurate water streamflow within the catchment (Bellin et al., 2016) but also considering the COSMO-CLM climate model for future projections. A wider range of inflow values driven by a set of climate models could provide a larger set of results that can be used for further stochastic analysis of turbined water and stored volume. Nevertheless, the climate model has been demonstrated to represent conditions in mountain regions well (Montesarchio et al., 2013) and different than other climate models, depicts general conditions of decreased precipitation over the catchment (Table 4). Hence it provides conservative information on possible impacts on streamflow and volume management. The results from the GEOTRANSF application assumed a conservative condition of upstream water use set at the maximum licensed withdrawal values. This information was kept unchanged for future scenarios, although possible variations in the future (e.g. from agricultural and touristic uses) may affect river water flows.

Moreover, the presented study considered precipitation, water flow and volume trends over a 30-year period considering their monthly average values, important for long-term large-reservoir management. For this reason, conditions of a high volume with a very short duration might have been potentially underrepresented.

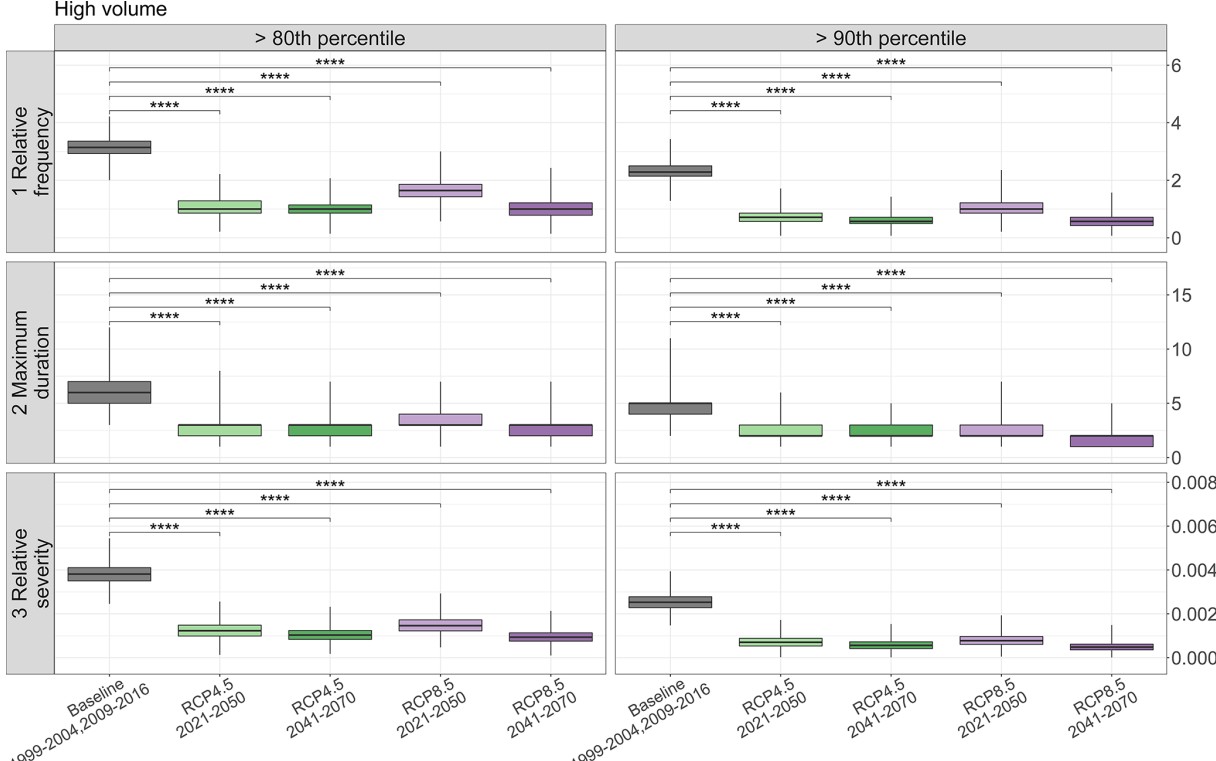

**Figure 9.** Boxplots of metrics of relative frequency, maximum duration and relative severity calculated from volume values greater than the 80th and 90th percentiles. The simulated volume values used for the metric calculation resulted from the Monte Carlo approach by randomly sampling from the volume prediction bands. The symbol **** refers to a $p$ value $\leq 0.0001$. For an interpretation of the references to colour in this figure legend, the reader is referred to the web version of this article.

The statistical models are an effective tool to replicate past observations of water volume and turbined water outflows. Applying such a regression to future conditions of predictors, reservoir management in terms of turbined outflow as well as minimum ecological flows was assumed to be stationary over time. Nevertheless, such a constraint is justified by the high uncertainty associated with future changes in hydropower production patterns affected by societal conditions (e.g. energy price fluctuations; Gaudard et al., 2014; Ranzani et al., 2018). Moreover, the minimum ecological flow was always set to the values established by law, although critical conditions in water availability (e.g. streamflow) may lead to extraordinary changes in minimum flow which can affect the stored volume as well as the turbined outflow.

Finally, the selected models considered a few tested and selected variables. Although other variables play important roles within the management of the reservoir at different temporal resolutions (e.g. hourly energy market price), the simulation on monthly aggregated values supported the objective of analysing long-term variations in the stored volume for the large S. Giustina reservoir.

## 6 Conclusions

The SDM represents the overall behaviour of the system in terms of turbined outflow (water demand) and future conditions of reduced water availability over a 30-year period well. Changes in water availability can deeply affect the actual turbined water, which plays a strategic role in the economy of the province, as in the whole Alpine region. Moreover, reduction in the water streamflow can have consequences in terms of ecological hazards and water supply quality downstream of the reservoir. The S. Giustina reservoir plays a crucial role in buffering water variations in the Noce catchment and downstream. Due to its size, type and position it is strategic for hydropower regulation and hydrologically disconnecting upstream with downstream river flow.

The modelling chain from climate change projections to the hydrological model water flow output and their use in the stochastic SDM provided to be a quick and effective tool to explore trend conditions of the S. Giustina reservoir volume and turbined outflows.

Results of both stored volume and turbined outflow suggest that even under RCP4.5 in the short-term scenario, reductions in terms of volume and turbined outflow will be severe with monthly average reductions for outflow and vol-

ume values respectively from April and May onwards and persisting throughout the year. This period of negative variations should be considered for the adoption of adaptation strategies focusing on water demand reduction. While considering months of expected increases in water availability as preparation periods, implementing strategies of earlier reservoir water accumulation is necessary while preparing for persistent conditions of lower availability compared to the baseline period. Such a strategy could prevent downstream conditions of water shortages during summer, autumn and part of winter while also preparing for reductions in turbined water for hydropower especially during summer and winter months of high electricity peak loads. Adaptation strategies should consider the results on generalized future conditions with an increase in frequency of months, maximum number of consecutive months and relative severity for all scenarios of low volume below the 10th and 20th percentiles. Consistently with these projections, frequency, duration and severity metrics for high-volume events below the 80th and 90th percentiles are expected to decrease. These results call for adaptation strategies of coordinated actions across those socio-economic sectors relying on abundant water demands (e.g. for agriculture) to face more frequent and longer periods of higher reduction of stored volume compared to the past.

Future model expansions will include water demand from multiple human activities (e.g. agriculture and domestic) and their effects on water availability reduction from upstream to downstream. By doing so, SDM models can support the understanding of criticality connected to unsustainable water demands and anticipate critical conditions, in order to inform dam managers and local authorities on the timing and importance of climate change adaptation strategies. Moreover, the use of open code and libraries for the assessment of variables' interactions through statistical models make SDM transferrable to other cases at the interregional/transnational scale in combination with available water flows datasets and open hydrological models (e.g. Copernicus, LISFLOOD model).

Finally, this analysis sheds light on the need to consider future changes in water availability and their consequences on already existing human activities relying on abundant water resources, thus being unprepared to quickly adapt to future climate impacts. Results should be considered in future plans to change S. Giustina management practices to reduce climate change impacts on reservoir operations. The findings presented reinforce the Alpine region's water tower vulnerability to supply water and ensure its use for power production. This is the first step for more comprehensive water scarcity assessments in order to provide policymakers with information in line with the European Water Framework Directive on potential adaptation strategies to gain systemic leverage effects on sustainable water management and climate change adaptation in the Alps (Alpine convention, 2013; European Commission, 2018, 2021).

*Code availability.* The source code for data processing and analysis developed in this study is freely available at https://github.com/ Ste-rzi/SGiustina_future_SDM (last access: 26 July 2021; Terzi, 2021).

*Data availability.* The data that support the findings of this study are available from the province of Trento – Agency for Water Resources and Energy, but restrictions apply to the availability of these data, which were used under license for the current study and so are not publicly available.

*Supplement.* The supplement related to this article is available online at: https://doi.org/10.5194/nhess-21-1-2021-supplement.

*Author contributions.* StT was involved in the project's conceptualization, data curation, formal analysis, methodology development, software usage, result validation and visualization and the preparation of the original draft of this paper. JS was involved in the project's conceptualization, formal analysis and methodology development and reviewing and editing of this paper. StS was involved in the project's visualization and supervision and reviewing and editing of this paper. SiT was involved in the project's visualization and supervision and reviewing and editing of this paper. AC was involved in the project's supervision and reviewing and editing of this paper.

*Competing interests.* The authors declare that they have no conflict of interest.

*Special issue statement.* This article is part of the special issue "Drought vulnerability, risk, and impact assessments: bridging the science-policy gap". It is not associated with a conference.

*Acknowledgements.* The authors acknowledge Eurac Research for supporting the research activities on multi-risk assessment in mountain regions within the context of climate change and the Provincia autonoma di Bolzano – Alto Adige Ripartizione Innovazione, Ricerca e Università as a financing institution of the AquaMount project. The authors also thank the Autonomous Province of Trento for data provision and Oscar Cainelli for making data and hydrological model output available and Stefan Steger and Mattia Rossi for R and statistical support.

*Financial support.* The research leading to these results has been partly funded under the PhD programme in Environmental Science of the Ca' Foscari University of Venice (PhD research

grant). Additional funding was provided by the Provincia autonoma di Bolzano – Alto Adige within the AquaMount project (grant no. D59C20000160003).

*Review statement.* This paper was edited by Gustavo Naumann and reviewed by two anonymous referees.

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
