# Peer review of "Stochastic System Dynamics Modelling for climate change water scarcity assessment on a reservoir in the Italian Alps"

_Natural Hazards and Earth System Sciences, 2021_

## Author Comment (AC1)

**Reply to referee 1**

We would like to thank you for your constructive comments and feedback on this manuscript. We think that the suggested revisions based on the Referee's comments will certainly improve the article. Please find our responses (in blue) to the main points raised (shown in black) below.

**Introduction**

The SDM methodology needs to be better introduced. Section 1 fulfill this role, but in my opinion it needs to be integrated directly in the introduction instead of being a separate section.

→ Thank you for your suggestion and we agree that bringing the current section 1 into the introduction would provide more information on SDM up in the manuscript while making it more synthetic. We will implement it in the revised version.

**Methodology**

This section needs to clearly highlight where the stochastic component of the model plays a role. If I have understood correctly, only the modeling of the reservoir level and outflow is done in SDM, whereas the input flow is modeled with a deterministic approach. Since human activities (and climate change) may influence also the upper basin, some considerations on the assumption made may be useful for a reader. E.g. How much anthropogenic activities are in the upper basin?

→ Yes, reservoir level and outflow modelling are modelled through regression models and analysed following a stochastic approach. We acknowledge we should bring up earlier in the methodology some of the assumptions on human activities that are now presented in the future projections and limitations sections (line 235 and 374) and we will implement these suggestions in the revised version.

**Results**

I found very limiting the lack of tests on the statistical significance of the differences from the baseline. This is particularly relevant since you are using just a single climate projection model.

Additionally, the analysis on the extremes (30th and 80th percentile) needs to be expanded. Limiting the analysis on the number of events may severely bias the analysis. You should integrate with the duration of such events, number of days under a threshold and the severity (in the case of drought).

It can happen that a smaller number of events in the future is due to the occurrence of less event but much longer.

→ Thanks for the constructive feedback. We see the need to introduce statistical tests and further expand the analysis of extreme conditions. We will integrate statistical tests for baseline vs future projections as well as integrating the number of events in the future with other metrics in the revised version, such as the maximum number of consecutive months below/above the selected thresholds (duration) and the changes in the number of events below/above different thresholds (magnitude).

**Specific Comments**

P2 L51-57. Here a more detailed explanation of SDM is need, with a clear description of the advantages and disadvantages of deterministic vs. stochastic approach (move and reword from Section 1).

→ Similarly to the first comment, we agree that bringing the current section 1 into the introduction would provide more information on SDM up in the manuscript and we will implement it in the revised version.

P3 L72-73. Can you please better clarify the role of connectors and the differences from converters? (maybe using two different examples rather than the same).

→ Yes, we can rephrase the sentence providing a different example, as follow: "converters - parameters influencing the flow rates (e.g. temperature variable acting to alter evaporation from a water body) and (iv) connectors – as arrows transferring information within the model (e.g. linking the monthly effects on reservoirs' water discharges) (Sterman et al., 2000)."

P4 L94-99. This paragraph would fit better in Section 3 in my opinion. In general, Section 1 seems unnecessary, and I will move its content in either introduction (first part) or Section 3 (second part).

→ We agree with your suggestion to move this paragraph in section 3.

P5 L125. It would be nice to have these numbers contextualized in comparison to e.g. average flow.

→ Thanks for the suggestion and we can report in brackets the percentage with reference to the average annual flow.

P6 L149. Fig. 2. It is not clear to me the role of soil (type?) and land-use as vulnerability factors. Please briefly clarify in the text.

→ Soil and land-use are reported in Fig2 for completeness and as background information coming from the GeoTransf hydrological model output. Both these issues were considered as vulnerability factors affecting run-off and its effect on the reservoir stored volume and water turbined. Although they were both kept constant in case of future conditions, we thought their representation could provide a wider picture on those factors affecting the final risk conditions. We can specify this role in the text (lines 145-146) of the revised version.

P6 L150. part of the caption in Fig. 2 seems missing.

→ Thanks for pointing that out and here reported the full caption that we will report in the revised version: "Causal loop diagram used to describe the risk variables and their interactions leading to critical states of S.Giustina reservoir operations. Climate variables are in green font, blue font for hydrological-related components, yellow font variables are those involved in S.Giustina operations leading to critical states".

P7 L158. Fig. 3. I understand the need to have different "baseline" period in different components. However, is there any way to homogenize those references or at least use a different name for each one of them? At the moment, when you refer to baseline in the text it is really difficult in some cases to understand to which period are you referring to.

→ We understand such difficulty, although the use of different baselines is constrained by the data availability. As suggested, we can explicitly add the variable name and its baseline period in the revised version in order to prevent potential misunderstanding.

P9 L197. Table 2. It is not 100% clear to me what are the other models reported here. Are those the best for each model type (among the ones in supplement materials)? Please clarify.

→ Yes, the model reported here are the best for each model type. The whole list of tested model are then reported in the supplement material. We agree that this can be clearer and we can improve the text and the table caption.

P9 L197. Table 2. Please highlight in the table the models selected as "best".

→ Thanks for the suggestion and we will make bold the selected model in the revised version in table 2.

P10 L216. Eq. (2). Is the random effect still just the month as in Eq. (1), or is it in the first term as well through V? Please clarify an eventually emphasize this key difference.

→ Thanks for pointing that out as in both equations the random effect is represented by the variable "month" and there is indeed a nested effect, which is due to the monthly effect on both inflow and volume variables. We acknowledge and agree on the need to emphasize the presence of nested models in the revised version of the text.

P10 L227. I'm not familiar with this procedure, but I'm assuming that you ends with 58 calibrations. which one is used at the end, or how are them combined? Please clarify.

→ Yes, all the 59 calibrations are used to compute a mean value of each model performance in predicting Volume and water turbined (R2 and RMSE). By doing so, we aimed to avoid common limitations of arbitrary choice of splitting a dataset into training and testing, which can provide biased estimation of model's performances. We agree that this needs to be clearer and we will specify the computation of a mean value from the forward time-window approach.

P11 L242. Why this asymmetric choice instead of 20 and 80? This needs to be justified.

→ Thanks for pointing that out. This assumption was based on previous studies reported in the text (Majone et al., 2016; Yilmaz et al., 2008) characterizing low and high flow in flow duration curves where the threshold of 0.2 exceedance probability was selected for high flows and the threshold of 0.7 exceedance probability was selected for low flows. Such assumption was reported here as the two corresponding thresholds of $30^{th}$ and $80^{th}$ percentiles to characterize critical conditions. Following your suggestion, we can integrate these thresholds adding the 10th and 90th percentiles in order to provide an additional description of critical conditions in a symmetric way.

P11 L244. Which baseline period? It may worth to specify here.

→ We agree that we need to specify in the text that we are referring to volume values from the baseline period (1999-2004 & 2009-2017). We will report this information in the revised version.

P11 L247. It is not clear to me where the Monte Carlo approach is used. Is it used only for the percentile analysis?

→ The Monte Carlo approach is used to analyse future values of volume stored below and above the selected $30^{th}$ and $80^{th}$ quantiles in order to provide a more robust assessment (i.e. having a larger set of future values based on the regression models) of critical conditions in the future. We agree this can be clarified in the current paragraph.

P11 L254. This sentence is a little confusing. I'm assuming that the storage for "flood prevention" is higher than the maximum due to the additional volume stored to prevent a potential downstream flood, but the way the sentence is presented is confusing. Please reword.

→ Thanks for pointing that out and we can rephrase it in the revised version into: "Figure 4 shows the modelled and real values, with volume ranging from 0 (i.e. no usable volume) and a maximum level of 151.20 Mm3. Only in case of emergency flood prevention additional volume can be stored up to 159.30 Mm3."

P12 L257. Figure 4. Please clarify the meaning of the dotted lines (percentiles?). Same in Fig. 5.

→ Thanks for pointing that out. We will integrate the use of dotted lines for percentiles in figure 4 and 5 in the revised version.

P12 L265. If I understand correctly, this means that the most important component (inflow) is the one that is not affected by the stochastic modeling. This needs to be further discussed and emphasized in order to clarify the value of using the stochastic approach in similar conditions.

→ Yes, inflow is used as an (deterministic) input variable to predict water turbined and volume stored. While deterministic assumptions of variables interactions are often

implemented to describe dynamics of a system, the use of statistical regressions allows to replicate past conditions, predict and stochastically test future ones accounting for its uncertainty range. We agree that this can be emphasized in the text and we can implement it in the revised version.

P13 L270. In the whole section, statistical significance of the changes need to be reported. Even changes with different sign may not be that different if the changes are not statistically significant (also accounting for natural fluctuation within the 30-year window). See Welch test or similar.

→ Thanks for the constructive feedback. We see the need to introduce statistical tests between baseline and future values and we can integrate them in the revised version.

P 14. Figure 6. I'm not sure that showing each single year is relevant, since these are projections. I would find a way to show time slices rather than annual/sub-annual values.

→ We see the need to better emphasize trends rather than single projected values. Following your suggestion, we will replace them with (e.g. 10-year average) time slices.

P15 L302. It would be interesting to integrate the analysis with the result in terms of contribute to the annual storage. Differences in rainy months may be much more relevant for the total storage that during the dry months, even if the percentage differences are comparable.

→ Thanks for bringing this in the discussion as in our initial discussions we computed Fig 7 and 9 rescaling them to the total annual water turbined and total annual storage (hence providing information on average month effect over the whole years). However, since these results have a similar trend to the monthly percentage variation figure (current fig 7 and 9), we opted to only provide this latter.

P16 L312. Number of events is not the only relevant metric. For drought/water scarcity: are those events longer? what about the number of deficit days per year? Is the total/average deficit of these events increasing? For flood: is the max surplus increasing? Those are examples of simple analyses that can be added without much effort.

→ Thank you for the suggestion, this is an interesting analysis that we can integrate into the revised version to improve the characterization of future critical conditions. For this purpose, we can compute the changes in the number of months below/above different thresholds (i.e. magnitude) as well as the maximum number of consecutive months below/above the selected thresholds (i.e. duration).

P16 L322. You should add also the same statistics for the reference period. How those number differ from the reference values? Are the differences in the short term (2021-2050) in line with what is already observed? Is the current condition (2021) very different from the reference period? Some insights on that would be useful to understand the reliability of these projections.

→ Following the previous comment, we can compute different metrics for the volume baseline period and future scenarios of RCP4.5 and 8.5 both 2021-2050 and 2041-2070 in order to explore trends and differences.

P17. L340. Please check the use of the term "negatively" here (and in the rest of the discussion as well). This can be interpreted as either mathematically (sign minus) or qualitatively (to worse). I suggest to rephrase.

→ Thanks for pointing that out. We agree and will rephrase it.

P18 L353-354. What about flood? Can increasing storage during winter be a problem in case of flood? Please elaborate.

→ Thanks for pointing out such an interesting side-effect due to the implementation of adaptation strategies such as an earlier water accumulation. We see the point of clarifying how such strategy can affect flood events. In particular, we will clarify that our discussion is based on:

- o The trend of volume decreases in spring and summer showing higher percentage values than the increases in winter (from figure 7)

- o The results on the number of events higher than the 80$^{th}$ quantile (Figure 8), which are expected to decrease for all scenarios except for the RCP8.5 2021-2050 (showing higher values than for the baseline volume).

- o The additional storage for "flood prevention", which availability needs to be always ensured and managed together with the Civil Protection to prevent a potential downstream flood, even in case strategies of earlier reservoir accumulation would be considered.

P18 L356. Increasing in high and low volumes? I'm not sure that this sentence is in line with your results. Please clarify.

→ Thanks, we will rephrase it in order to refer to the number of events with volume below than the 30$^{th}$ percentile and above the 80$^{th}$ percentile.

P18 L359. I think that the Monte Carlo results may give an idea on the robustness of such changes, but this is not discussed at all at the moment.

→ Thanks, we see the need to bring the Monte Carlo results into the discussion. We can integrate this into the revised version.

P18 L364-365. This is more a conclusion that a discussion of the results.

→ Thanks for pointing that out and we will move such a sentence into the discussion section in the revised version.

P18 L375. This is a very important point that needs to be highlighted in the methodology as well.

→ Thanks and we agree that having such information earlier in the methodology can help readers to better understand such an assumption on human activities.

---

## Author Comment (AC2)

**Reply to referee 2**

We would like to thank you for the detailed and valuable feedback on this manuscript. We think that the suggested revisions based on the Referee's comments will certainly improve the article. Please find our responses (in blue) to the main points raised (shown in black) below.

**GENERAL COMMENTS**

A. The authors state that an innovative stochastic System Dynamics Modelling (SDM) approach is applied to simulate water stored and turbinated in the S. Giustina reservoir under current/past and projected climate conditions (p 1, l 18-20). SDM is described as "an approach used in the field of complex system behaviour" useful for analysing large systems and interactions between subsystems (p 3, l 69-79). In presenting the case study, emphasis is put on the importance of achieving "a better understanding of the complex interactions in the S.Giustina water management", and it is stated that "SDM can help to depict the S.Giustina reservoir dynamics and its responses to future pressures, including climate change and anthropogenic factors" (p 5, l 127-132). However, the SDM approach simulates reservoir volume and outflow using reservoir inflow produced by the GeoTransf hydrologic model as input (Fig. 3 box 3, p 8, l 168, and l 181-184). Its implementation consists of two regression models simulating reservoir volume as a function of inflow and month (Eq. 1) and outflow as a function of inflow, past volume and month (Eq. 2). This simple approach seems not to correspond to the provided SDM definitions.

→ Thanks, we see the need to better match the description and expectations in the initial part with the actual implementation. Indeed our application tackles one portion of multiple issues related to water management and scarcity. However, we also recognize the need to provide a wider context and reference to other existing SDM applications incorporating several modular components and leading to (more) complex systems analysis (Duran-Encalada et al., 2017; Gohari et al., 2017; Sušnik et al., 2018, Malard et al., 2017; Stave, 2010; Davies and Simonovic, 2011). In the revised version, we can soften some of the initial expectations, while referring to novel SDM applications and its possible modular integration in a wider modelling context.

B. The authors argue that the novelty of their approach lies in the application of SDM in a probabilistic/stochastic framework, in order to account for the uncertainties of future climate scenarios (p 2, l 51-57; p 4, l 94-95; and p 8, l 174-176). However, the stochastic (Monte Carlo) approach is given a very short and unclear description at the end of Sect. 3 "Material and methods" (see specific comment 21). Moreover in the results section, the outcomes of the stochastic sampling shown in Fig. 6 and 8 are either not discussed or only mentioned briefly (see general comments G and H).

→ Thank you for pointing that out. Similarly to the comment provided by Reviewer 1, we acknowledge the need of improving the description of the MonteCarlo approach in Section 3 as well as expand the results coming from its application. We will implement the suggestion by further describing in more detail the Monte-Carlo/stochastic procedure in the revised version.

C. The role of anthropogenic pressures is mentioned in several parts of the manuscript (e.g. p 2, l 61; p 4-5, l 108-116). The use of water demand/withdrawal data is mentioned briefly only on p 11, l 235 ("GeoTransf simulations considered unchanged maximum water withdrawals in the Noce catchment") and on p 18, l 374 (similar statement). However, these data are not described nor referenced in the materials and methods section. Most importantly, their role in the modelling framework is not described at all. The conclusions read that "Future model expansions include water demand from multiple human activities" (p 19, l 401-404). The authors should clarify whether non-hydropower water abstractions are simulated and, if applicable, describe how these are included in the modelling framework.

→ Thanks for your comment. We see the need of improving the reference to the conservative assumption of non-hydropower water abstractions upstream of the reservoir considered at their maximum value (according to the provincial licensed withdrawals, which data is not publicly available). We can add this information in the material and methods section referring to it within the text and adding references in the modelling framework section. In general, the assumption of constant maximum withdrawals included in the GeoTransf simulations provided a conservative estimate for the aim of looking at changes in the S.Giustina volume and water turbined due to water flow variations from climate change effects.

D. SDM components (stocks, flows, converters, connectors) are introduced in Sect. 1. However, the terms "stock", "converter" and "connector" do not appear in the rest of the manuscript. These terms should be used when describing the methodology (throughout Sect. 3 and in particular in Fig. 2) to clarify why SDM is relevant for this study. Moreover, the definitions and explanatory examples of "converters" and "connectors" are unclear (p 3, l 71-73). Evaporation rates, which are used as examples of "converters", may also fulfil the definition of "flows" ("variable's rate of change"). Instead, should the parameters of evaporation equations be regarded as "converters"? Otherwise, are "flows" fluxes of material between "stocks" simulated by the system (e.g. reservoirs and rivers) rather than fluxes leaving the system (e.g. evapotranspiration)? The example of "linking of temperature variations to evaporation rate" may indicate that "connectors" act in the opposite direction of "converters", but it is not clear how these two categories are qualitatively different. These definitions should be clarified, especially because the book by Sterman et al. (2000) is not an open-access publication.

→ Thanks for pointing that out. We will add reference to the SDM terms in figure 3, where they have been reported, and in the text describing figure 3, especially in the last part focusing on box #3 (l 174-180). The comment on the use of explanatory examples was also raised by Reviewer #1 and we proposed a different example: "converters - parameters influencing the flow rates (e.g. temperature variable acting to alter evaporation from a water body) and (iv) connectors – as arrows transferring information within the model (e.g. linking the monthly effects on reservoirs' water discharges) (Sterman et al., 2000)." We trust the new text is clearer in its meaning of the terms, offering better distinction between flows and converters.

E. The description of the "causal loop diagram" (Sect. 3.1) is too brief and needs to be rewritten as the roles of several elements described in the text and shown in Fig. 2 are not understandable. In general, the authors should clarify to which extent the arrows in the diagram

correspond to the flow of information between models shown in Fig. 3: if these representations overlap, then Sect. 3.1 and 3.2. could be merged into a single section. More specifically, in what ways do "soil" and "land-use" represent system vulnerability? Also, it is not clear how the "month of the year", "water turbined" and "reservoir volume" represent exposure. Finally, the "critical states" are undefined.

→ Thanks for raising this point. We see the need to provide additional information on the factors in the Fig2 description. However, we also recognise its importance as a first expert-based conceptualization feeding the following model set up as it is also carried out in other SDM applications. For this reason, we will integrate its description and relation to Fig.3.

F. In Table 3 (Results), average outflows are between 8% and 9% smaller than average inflows. These discrepancies, which are not discussed in the manuscript, may imply (i) that the closure of the reservoir water balance is not guaranteed, as over long time periods total inflow and outflow should be very close (except a change in storage that would be very small compared to the cumulated flows); or (ii) that the difference between inflow and outflow is lost via evaporation, percolation, non-turbinated releases, or abstracted for other uses than hydropower. However, none of the latter losses/uses are mentioned in the manuscript. The combined use of Eq. 1 and 2 (or the corresponding models #3 and #4 in Table 2) to model volume V and outflow Qout may violate the reservoir water balance: once Qout(t) is calculated with Eq. 1, as V(t-1) and inflow Qin(t) are also known from previous calculation (or initialisation) and input data, then V(t) should be obtained by applying the reservoir mass balance: V(t) = V(t-1) + Qin(t) - Qout(t), where V(t) is the volume at the end of month t, and Qin(t) and Qout(t) are average inflow and outflow during month t. The authors should clarify how the closure of the reservoir water balance is guaranteed, as this is a prerequisite for a reservoir simulation model: to do so, they may replace the expressions in Table 2 with understandable mathematical equations, and define the "f" functions in Eq. 1 and 2 (see specific comments 17-19). If the water balance closure is not guaranteed by this approach, the reservoir simulation model should be revised.

→ Thanks for raising this concern. The use of the two regression models for volume stored and turbined water accounted for the water balance equation variables and dependencies on water inflow and volume from the previous time step. This provided a better fitting of both variables on past observations compared to the use of the predicted turbined water into the water balance equation to calculate the volume stored. However, we recognize the need of satisfying the water balance at each time step. We can implement it to simulate the S.Giustina volume (although having a lower performance to replicate past values) and correctly report it in the revised version of this manuscript.

G. The presentation of results needs to be improved. For example, Fig. 6 is not discussed in the text: it is referenced only once on p 13, l 272 while discussing the average values presented in Table 3. Figure 6 contains 12 time series plots and seems to use the grey shaded area to show stochasticity, which is presented as a major component of the methodology. Therefore, Fig. 6 should be discussed extensively, focusing in particular on the value of the information provided by the stochastic approach for the water scarcity risk assessment. Moreover, the confidence interval is only mentioned in the caption of Fig. 6 without specifying the confidence level nor the

assumptions underlying its calculation, thus hindering the interpretation of the plots. Also, the apparently smoothed lines in volume and outflow plots (Fig. 6) are not defined nor discussed in any part of the manuscript. For other examples, see specific comments 22-27.

→ Thanks for the feedback. We acknowledge the need to better describe and present the results covering all the reported information. In agreement with the comment from Reviewer #1, we can integrate Figure 6 with better caption information and modify the figure in order to replace single projected values with aggregated time slices.

H. The emphasis on the probabilistic/stochastic framework is not followed by a thorough discussion of the uncertainty intervals derived from the Monte Carlo sampling. For example, Fig. 7 and 9 only show average variations. Moreover, although some outcomes of the stochastic approach are shown in Fig. 6 (grey areas) and 8 (box-plots), these are either not discussed at all (Fig. 6) or mentioned only briefly (p 16, l 310-315) in the manuscript.

→ We acknowledge the need to improve the discussion on the MonteCarlo sampling application with reference to Figure 8 as this was also pointed out by Reviewer #1. While Figure 7 and 9 reported averaged conditions on a monthly basis, but are not directly connected to the MonteCarlo sampling, Figure 8 aims to provide information on the results of the MonteCarlo sampling and we can further discuss it in the revised version.

**SPECIFIC COMMENTS**

1. p 1, l 20: Based on the information provided in Sect. 3.2, the sentence "The integration of outputs from climate change simulations as well as from a hydrological model and statistical models into the SDM is a quick and effective tool to simulate past and future water availability and demand conditions." should be replaced by a more informative statement on how the components of the model tool-chain are assembled, e.g. explaining briefly the input/output flow between model components.

    → Thanks and we can briefly clarify the model chain statement.

2. p 1, l 21-23: Add simulation periods to "short-term", "long-term" and "baseline".

    → Thanks for pointing that out. We agree and will add them.

3. p 1, l 25 and throughout the text: Replace "quantiles" with "percentiles" when using values within 1 and 100.

    → Thanks and we will modify them throughout the text.

4. p 1, l 27-29: This statement seems not to be supported by the results, as no other water-demanding sector than hydropower is considered in this study. Moreover, the impact on hydropower production is not quantified, as model output includes reservoir storage volume and outflow but not energy production.

→ Thanks for this comment, but this last part of the abstract does not claim to be supported by results. We will rephrase the statement to make this distinction clearer since it provides an outlook on possible future upscaling and application of SDM to include other water demanding sectors and further characterize water scarcity conditions.

5. p 3, l 89: If possible a scientific publication should be referenced when asserting the "reduced accuracy in comparison with dedicated physically based models".

→ Thanks, we can rephrase the sentence to clarify that SDMs are not meant as replacements for dedicated physically-based models in a given field, but instead are useful in exploring and connecting disparate fields (e.g. Haung et al, 2011).

6. p 4, l 101-105: As detailed data regarding reservoir storage and environmental flow requirements are provided later (p 5, l 126), adding the following information about the Noce river would help understanding the context: catchment area, average discharge, and sectoral or aggregate water demand/abstraction volumes. In particular, water demand/abstraction information would be of crucial importance in light of the mentioned water scarcity events/issues (p 5, l 108-116) and because the S. Giustina reservoir provides water for irrigation and other downstream uses (p 5, l 122-124).

→ Thanks, we can add information on the catchment area (1367 km$^2$) and the aggregated sectorial demand for the whole catchment as reported in the following table (Percentages subdivision of the licensed water withdrawals per sector (year 2017) where large hydropower plants (with average nominal capacity greater than 3 MW) water withdrawals are excluded (Data source: Provincial Agency for Water and Energy)):

| Sector | Percentage [%] |
|---|---|
| Agriculture | 16.39 |
| Other | 0.15 |
| Domestic | 3.98 |
| Hydropower | 77.81 |
| Industrial | 0.49 |
| Snowmaking | 0.28 |
| Fish farming | 0.90 |

The high percentage value of 77.81% of licensed withdrawals for hydropower only accounts for the small run-of-the-river plants, hence providing information on the very high use of water for hydropower purposes within the Noce catchment where the S.Giustina dam is the largest reservoir.

7. p 5, l 124-126: The statement on stakeholder concerns about "unused" water releases should be supported with some evidence. Otherwise it should be removed.

→ Thank you for the feedback and this is a critical point we agree should only be kept if supported by references.

8. Figure 2: Clarify whether precipitation is partitioned into snowfall and rainfall as a function of temperature. If so, then rainfall should be added to the diagram. Consider whether the distinction between "Climate-related hazards" and "Hydrological-related hazards" is necessary. The caption seems to be truncated.

→ Thanks, we can integrate the caption and add rainfall to the CLD as precipitation was partitioned into snowfall and rainfall in the GeoTransf application.

9. Figure 3, legend: The terms "external component" and "internal component" are not discussed in the text: explain with respect to which other elements they are to be considered external/internal. Also, clarify the meaning of "Field/sector" and whether its graphical identifying feature is the dashed line or the fill colour.

→Thanks for pointing that out. We can add an explanation of the external and internal components division in the 3.2 section in order to clarify those components that were already developed by others (external) and those that were internally developed (internal). Also, "Field/sector" are represented by the combination of dashed line and fill colour.

10. Figure 3 and p 7, l 160-161: Provide the source of weather station data. Also, describe briefly the bias correction methodology, as "Quantile mapping" is mentioned in Fig. 3 but not in the text.

→ Thanks and we can add reference to the weather station source, as it was provided by the Provincial Met Office (Province of Trento – Weather service https://www.meteotrentino.it/index.html#!/content?menuItemDesktop=111). We can mention the bias correction in the text and add references to the GeoTransf application, since it belongs to the external component.

11. p 7, 8 and Fig. 3: The baseline simulation period needs to be clarified. From Fig. 3 and p 7, l 160-161, it seems that weather station data (period 1971-2005) are used to bias-correct 1975-2005 COSMO-CLM precipitation and temperature (l 177), which are in turn used as GeoTransf input. However, Table 1 reports that "Geotransf_inflows" cover the period 1981-2010, while in Fig. 3 the hydrological model baseline period is 1980-2010 and the SDM baseline periods are 1999-2004 and 2009-2017. A more detailed description of the input/output flow between model components could clarify these apparent inconsistencies.

→Thanks for pointing that out and there is indeed an oversight. COSMO-CLM (1971-2005) data were bias corrected with observed data (1981-2005) and used in GeoTransf to replicate past stream flows during the Orientgate project. However, due to the missing temporal overlap of GeoTransf values with the S.Giustina turbined water and volume stored

values, the regression models considered the 1999-2004 and 2009-2017 baseline accounting for the observed streamflow during that period. We see the need of integrating such explanations within the text, correcting Table 1 and Figure 3 accordingly.

12. p 8, l 165-180: To understand the application of the GeoTransf model, further details are needed about the input data. In Fig. 3, glacier extension, land use and soil data seem to be model input, but they are either not mentioned or not explained in the text. Add the sources of these datasets. Clarify in particular whether glacier extension is an input or an output variable, as the following sentence seems to imply that glacier state is a model output: "GeoTransf provides a description of the hydrological dynamics within the Noce alpine river catchment, assessing variations in water contributions coming from climate change effects in terms of temperature, soil moisture, glaciers, snow and rainfall" (p 8, l 170-172).

→Thanks and we can improve the description of the input data by adding reference to the GeoTransf application, since it belongs to the external component as reported in Fig.3. Moreover, we can clarify the sentence within the text of the revised version of the manuscript.

13. p 7, 8: Add the temporal resolution(s) of input data and model output.

→ Thanks and we can add it.

14. p 8, l 166-168: Clarify what is meant by "these blocks" (temperature and precipitation data? GeoTransf output?) and provide references in addition to the OrientGate project website. Check if the latter should be updated to http://m.orientgateproject.org/.

→We can replace "These blocks" with "these boxes" in order to create consistency with the used terms within the text, hence referring to box 1 (climate projection) and box 2 (hydrological model). Thanks for pointing at the new website and we can certainly use it to replace the previous one.

15. p 8, l 182-184: Clarify this sentence. The meaning of "covers" is unclear: are reservoir volume and outflow outputs of SDM? What are the "critical conditions" to which volume and outflow are exposed?

→Yes, we can clarify that reservoir volume and outflow are outputs of the SDM. Critical conditions refer to both trends of increasing/decreasing water turbined and volume stored as well as volume conditions overcoming certain thresholds for (above the 80th percentile and below the 30th percentile from the baseline, but also 10th and 90th percentiles too as suggested in response to reviewer #1).

16. p 9, l 187-189: The following input variables are mentioned here for the first time: "hydroelectric energy market price", "water outflows from an upstream dam reservoir". They should be introduced in Sect. 3.2 ("System dynamics modelling set-up and input data") and integrated into the Fig. 3 flow chart. Now they are mentioned briefly in the Supplement without

any further detail than units, temporal coverage and source (missing for the upstream dam reservoir outflow).

→Thanks and we agree mentioning them earlier in the methodology section can clarify their use.

17. Table 2: Replace the formulas with mathematical expressions. The current expressions are not understandable. The reader should not be required to learn the R syntax in order to understand what seems to be regression models. This notation may lead to ambiguous interpretations of, for instance, the "lag" operator (how long is the lag?), the "1|month" expression, the "s" function, and so on. Moreover, model parameters are visible in these formulas. The same applies for Table S2 and S3 in the Supplement.

→ Thank you for the comment. We intended to show the R syntax to enhance transparency and to allow the reader to easily reproduce the general workflow within the R environment. However, we agree that adding the underlying mathematical expressions of the statistical methods further enhances traceability and therefore we can add them to table 2. Associated with this table, we can add an explanation of the components to the Table caption (e.g. "s" = Function used in definition of smooth terms within the gam model formulae).

18. Section 3.3.1: Equation (1) seems to suggest that the reservoir volume at monthly time step t is function of inflow at t and month of the year. Why is not V(t) function of V(t-1) as well? This may seem an incorrect representation of the reservoir water balance and should be clarified. Function "f" is not defined, which makes Equation (1) not fully understandable. Also, in the following sentence the expression "grouping effect" should be clarified: "As a random effect, the month of the year was selected (month) for its grouping effect on the recurrent water volume variations on a monthly scale".

→ According to the response to the general comment F, we will substitute equation #1 to correctly agree with the water mass balance equation.

19. Section 3.3.2: Function "f" is undefined, making Eq. (2) not understandable. Moreover, it seems that a flow (Qin) and a volume (V) are summed together, thus potentially violating the dimensional consistency of the equation.

→ Similarly to response #17, we agree to add the mathematical formula for clarity. While, the represented sum belongs to the linear mixed effect model syntax, we can briefly clarify the mean of grouping effect in the regressions in the revised version of the manuscript.

20. Section 3.4: The authors need to state explicitly which model parameters are calibrated. The description of the "forward time-window approach" needs to be improved. It seems that the whole procedure consists of a sequence of 59 successive model calibrations, each based on an increasingly longer training dataset. However, it is unclear how information from iteration n-1 is used in iteration n. Also, what algorithm is used to search the parameter space?

→ Thanks for the feedback. Calibration and validation procedure considered the best fixed-effect and random-effect coefficients in order to best predict reservoir volume and turbined water. Differently from physically-based models, there is no a priori information on the parameter space definition since this is directly derived from the available input data parameter range. Finally, all the information coming from each iteration on model performance is used to compute an average value of RMSE. We will improve the description of this process in the revised manuscript.

21. p 11, l 246-249: The description of the Monte Carlo approach needs to be clarified. The authors should clarify how the sampled volume time series are built, explaining in particular how the relevant characteristics of simulated volume are preserved (e.g. autocorrelation and seasonality).

→Thanks and similarly to the general comment H and the comment from Reviewer #1, we will improve the description of the MonteCarlo sampling providing more information on the sampling approach (time series of continuous 14 years as for the forward time-window approach).

22. p 11, l 255 and Fig. 4: Correcting simulated volume values to the maximum storage hides model overestimations. Moreover, the time steps of these corrections are not shown in Fig. 4, thus not allowing the reader to distinguish between simulated and corrected values. For transparency and to allow the evaluation of the modelling approach, volume values greater than the maximum allowed for flood should be shown in Fig. 4.

→ Although model overestimations were very limited in number, we can highlight those cases of corrected value in the revised version of the manuscript.

23. p 12, l 263-266: The sentence "[...] low values of reservoir volume compared to the monthly inflow values" seems incorrect: the active storage volume is 152.4 Mm3 (p 5, l 121) and the average baseline monthly inflow is 71.38 Mm3/month (Table 3). Moreover, if storage were small compared to monthly inflow, then the analysis should be carried out using a sub-monthly time scale (e.g. daily), as the reservoir would not be able to significantly regulate the flow at monthly scale.

→Thanks for the feedback. Regarding the concern related to the time scale for the analysis, we intended to provide information on the use of a monthly time scale resulting in a good prediction performance. However, we understand the need to make the sentence clearer and we can rephrase it in the revised version of the manuscript.

24. Figure 5: What are the horizontal dashed lines?

→Horizontal dashed lines represent the two selected thresholds for the 30th and 80th percentiles. We will integrate this description into the captions.

25. p 13, l 273-275: 2021-2050 RCP8.5 average inflow is 7.5% smaller than the baseline, while precipitation is 1.4% larger (Table 3). Is this due to increased evapotranspiration caused by the

relatively large temperature increase? The authors should discuss the impact of increasing temperatures on local hydrology.

→Thanks for the comment and we can integrate the results in Table3 with more specific discussions.

26. Table 3: the "Δ [%]" values seem wrong in several cases, e.g. the relative difference between 2021-2050 RCP4.5 and baseline temperature is 27.6% and not 0.5%. Similar errors occur for all other temperature changes. See also p 15, l 289.

→Thanks for pointing this out and we will modify the incorrect values.

27. p 17-18, l 344-346: Define "slow onset conditions of water availability" in the context of the presented results.

→Thanks and we can integrate this term with a brief description. We here refer to slow onset conditions as trends of increasingly reductions in water availability (e.g. slow onset usually associated with drought conditions) as projected by the climate projections and affecting turbined water and reservoir volume.

28. p 18, l 356-357: The sentence "At the same time, high volume events decrease [..]" seems to be redundant with the previous one, i.e. "[...] increasing number of future water scarcity conditions of high and low volumes stored".

→Thanks for the suggestion and we will make the sentence consistent with the previous one.

29. p 18, l 369: The sentence "Accounting for the GeoTransf application means relying on a very accurate water evaluation within the catchment" should be supported with some evidence/references. Moreover, the expression "water evaluation" seems ambiguous: the authors should mention what specific characteristics of the water system are reproduced accurately (e.g. river discharge?).

→Thanks for pointing that out and we can indeed clarify the simulation of water streamflows from the hydrological model.

30. p 18, l 376-377: Explain how the precipitation data used to force the GeoTransf model may miss intense precipitation episodes. This is not clear from the presentations of input data and results.

→The sentence in l 376-377 aims to highlight the focus of this study on trend variations at monthly scale on short- and long-term which can affect the S.Giustina water turbined and volume stored (e.g. Fig 7 and 9). We can clarify the sentence with more information on the GeoTransf model use and ability to replicate intense precipitation events providing reference to Bellin et al.,(2016) where full details of these procedures are carried out and described.

31. p 19, l 378: It seems that the authors refer to the reservoir volume time series length (14 years) as a factor that limits the model predictive performance. Model performance was evaluated using RMSE and R2 (Sect. 4.1). However, it is not clear how time series length negatively affects the goodness-of-fit of the regression models (Table 2). In contrast, small training samples may lead to over-fitting, which then can cause poor model ability to simulate time periods over which the parameters were not calibrated.

→Thanks and we actually reported this under "limitations" as we wanted to point at the possible negative effects of having limited training and testing samples due to over-fitting.

32. p 19, l 385-387: Was the monthly time resolution chosen because of the lack of sub-monthly (e.g. daily) data or following other considerations (as it seems from the brief mention on p 9, l 193-195)? This choice should be discussed in detail in the materials and methods section as it has a significant impact on the modelling framework, e.g. the authors have not used the energy price as a regressor when modelling outflow.

→ The monthly time resolution was selected since a daily resolution provided larger prediction error and monthly resolution better supported the trend analysis with information at the seasonal level.

References:
Bellin, A., Majone, B., Cainelli, O., Alberici, D., Villa, F., 2016. A continuous coupled hydrological and water resources management model. Environ. Model. Softw. 75, 176–192. https://doi.org/10.1016/j.envsoft.2015.10.013

Duran-Encalada, J.A., Paucar-Caceres, A., Bandala, E.R., Wright, G.H., 2017. The impact of global climate change on water quantity and quality: A system dynamics approach to the US-Mexican transborder region. Eur. J. Oper. Res. 256, 567–581. https://doi.org/10.1016/j.ejor.2016.06.016

Gohari, A., Mirchi, A., Madani, K.D.E. of C.C.A.S. for W.R.M. in C.I., 2017. System Dynamics Evaluation of Climate Change Adaptation Strategies for Water Resources Management in Central Iran. Water Resour. Manag. 31, 1413–1434. https://doi.org/10.1007/s11269-017-1575-z

Huang M., Elshorbagy A., Barbour SL., Zettle J., Si BC. 2011. System dynamics modeling of infiltration and drainage in layered coarse soil. Canadian Journal of Soil Science. 91: 185-197. DOI: 10.4141/cjss10009

Malard, J.J., Inam, A., Hassanzadeh, E., Adamowski, J., Tuy, H.A., Melgar-Quiñonez, H., 2017. Development of a software tool for rapid, reproducible, and stakeholder-friendly dynamic coupling of system dynamics and physically-based models. Environ. Model. Softw. 96, 410–420. https://doi.org/10.1016/j.envsoft.2017.06.053

Stave, K., 2010. Participatory system dynamics modeling for sustainable environmental management: Observations from four cases. Sustainability 2, 2762–2784. https://doi.org/10.3390/su2092762

Sterman, J.D., Burr Ridge, B., Dubuque, I., Madison, I., New York San Francisco St Louis Bangkok Bogota Caracas Lisbon London Madrid Mexico City Milan New Delhi Seoul Singapore Sydney Taipei Toronto, W., 2000. Business Dynamics Systems Thinking and Modeling for a Complex World.

Sušnik, J., Chew, C., Domingo, X., Mereu, S., Trabucco, A., Evans, B., Vamvakeridou-Lyroudia, L., Savić, D.A., Laspidou, C., Brouwer, F., 2018. Multi-stakeholder development of a serious game to explore the water-energy-food-land-climate nexus: The SIM4NEXUS approach. Water (Switzerland) 10. https://doi.org/10.3390/w10020139

---

## Author Comment (AC3)

Dear Mario Wetzel,

Thank you for your comments to the open discussion. Please find our responses (in blue) to the main points raised (shown in black) below.

1.  Figure 2: The CLD in figure 2 shows in a very comprehensive way the interlinkages of different factors that finally lead to the specific risk. However, the figure does not contain feedback loops. Could you please clarify the difference here to an Impact Chain (cause-and-effect chain) or a Directed Acyclic Graph such as used in Bayesian Network modelling. Did you consider and model direct feedback loops in your study?

The causal loop diagram (CLD) was developed as an initial conceptual framework and feedback loops were not identified in this specific case. It has indeed common points to the so-called Impact Chains (e.g. reference to the IPCC AR5 risk components) or to the Bayesian Networks (e.g. focus on variable's probabilistic representations) although system dynamics modelling aimed to characterize the time-dependent functions affecting the variables of the system under study.

2.  P. 2, 54ff: The manuscript mentions the benefits of the methodology to better understand the dynamics between anthropogenic and environmental processes. Looking at figure 2, I associate land-use, water turbined and reservoir volume to factors that are driven by anthropogenic processes and the interplay of anthropogenic and environmental processes. However, pertaining to exposure, "water turbines" and "reservoir volume" are solely driven by "month of the year". "Month of the year" remains a black box, in a sense that drivers that specify the differentiated role of intra-annual dynamics are not further specified. Is "month of the year" solely driven by precipitation variations, or also informed by differentiated economic activities, water consumption etc.?

Thanks and we see the need to better explain the variables under exposure as mentioned by the reviewers. While "water turbines" and "reservoir volumes" represent the element exposed to potential risk conditions, "month of the year" is here considered as a factor affecting the level of exposure and it is mostly related to the reservoir monthly management both due to the environmental (e.g. precipitation) and anthropogenic conditions (e.g. lowering the level of the reservoir in March-April to anticipate higher water inflow in May and June) from the previous years (baseline).

3.  P. 19, 383ff: Can the application of scenarios regarding the reservoir management be integrated here, or an alternative assumption (compared to the stationary) based on potential societal changes?

Thanks for this comment. The statistical approaches allowed to replicate past reservoir management and to keep them unchanged in the future for investigating potential critical states conditions of water turbined and reservoir volume. However, we see the possibility to play with future conditions of water inflow to the reservoir according to different scenarios of upstream anthropogenic water withdrawals since the upstream-downstream water management is an important feature to potential future disputes.

---

## Author Response (AR1)

**Author's response**

We would like to thank both Referees for their constructive comments and feedback on this manuscript. We think that the suggested revisions based on the Referee's comments has certainly improved the article. Please find our responses (in blue) to the main points raised (shown in black) below. Here reported the main improvements:

- We revised the reservoir simulation model in order to ensure the closure of the water balance equation at each time step.
- We integrated non-parametric tests throughout the analysis to evaluate significance of the differences from the baseline for both turbined outflow and stored volume.
- We integrated different metrics (accounting for frequency, duration and severity of low- and high-volume conditions) to better characterize the analysis on future critical conditions for the water stored in the reservoir.
- The stochastic implementation of SDM has been further described and discussed in order to provide relevant methodological and results information.

**Reply to referee 1**

**Introduction**

The SDM methodology needs to be better introduced. Section 1 fulfill this role, but in my opinion it needs to be integrated directly in the introduction instead of being a separate section.

→ Thank you for your suggestion, we moved the SDM description in the introduction accordingly.

**Methodology**

This section needs to clearly highlight where the stochastic component of the model plays a role. If I have understood correctly, only the modeling of the reservoir level and outflow is done in SDM, whereas the input flow is modeled with a deterministic approach. Since human activities (and climate change) may influence also the upper basin, some considerations on the assumption made may be useful for a reader. E.g. How much anthropogenic activities are in the upper basin?

→ In agreement with the comments from Referee 2, we integrated background information on the anthropogenic demands in the basin in the methodology, we modified Figure 3 to include the new reservoir simulation model represented through the SDM components and we further elaborated the application of the Monte Carlo approach.

**Results**

I found very limiting the lack of tests on the statistical significance of the differences from the baseline. This is particularly relevant since you are using just a single climate projection model.

Additionally, the analysis on the extremes (30th and 80th percentile) needs to be expanded. Limiting the analysis on the number of events may severely bias the analysis. You should integrate with the duration of such events, number of days under a threshold and the severity (in the case of drought).

It can happen that a smaller number of events in the future is due to the occurrence of less event but much longer.

→ We introduced a non-parametric statistical test comparing future projections with baseline conditions. Moreover, we introduced other metrics to characterize critical conditions of low and high volume stored in terms of:
   - duration (maximum number of consecutive months below/above the selected thresholds) and
   - absolute and relative frequency (changes in overall number of months below/above different thresholds and changes in the number of months below/above different thresholds over 14 years' time-window),
   - severity (average annual volume below/above the selected thresholds over the total volume stored over the 14 years' time-window) both for the baseline and future projections.

**Specific Comments**

P2 L51-57. Here a more detailed explanation of SDM is need, with a clear description of the advantages and disadvantages of deterministic vs. stochastic approach (move and reword from Section 1).

   → Thank you for your suggestion and we moved and reworded the SDM description in the introduction.

P3 L72-73. Can you please better clarify the role of connectors and the differences from converters? (maybe using two different examples rather than the same).

   → Yes, we rephrased the sentence in the revised version (line 70-75 in the marked-up manuscript).

P4 L94-99. This paragraph would fit better in Section 3 in my opinion. In general, Section 1 seems unnecessary, and I will move its content in either introduction (first part) or Section 3 (second part).

   → Thank you for your suggestion and we moved the SDM description in the introduction (double underlined green part in the marked-up manuscript).

P5 L125. It would be nice to have these numbers contextualized in comparison to e.g. average flow.

   → We reported in brackets that the minimum ecological flow is the only water flow downstream the dam (lines 178-180 the marked-up manuscript).

P6 L149. Fig. 2. It is not clear to me the role of soil (type?) and land-use as vulnerability factors. Please briefly clarify in the text.

→ After having received Referee 2 comments, we decided to remove Figure 2 due to its overlaps with Figure 3 and to give more space to the changes in the methodology section (e.g. water balance equation, different metrices of low and high volume stored).

P6 L150. part of the caption in Fig. 2 seems missing.

→ Please, refer to the previous comment.

P7 L158. Fig. 3. I understand the need to have different "baseline" period in different components. However, is there any way to homogenize those references or at least use a different name for each one of them? At the moment, when you refer to baseline in the text it is really difficult in some cases to understand to which period are you referring to.

→ We understand such difficulty, although the use of different baselines is constrained by the data availability. We explicitly added the variable name and its baseline period in the revised version in order to prevent potential misunderstanding.

P9 L197. Table 2. It is not 100% clear to me what are the other models reported here. Are those the best for each model type (among the ones in supplement materials)? Please clarify.

→ We modified the text to explicitly say that these are the best models for each model type referring to the supplement material for the whole list of tested models.

P9 L197. Table 2. Please highlight in the table the models selected as "best".

→ We highlighted it in bold within the table and mentioning it in the text.

P10 L216. Eq. (2). Is the random effect still just the month as in Eq. (1), or is it in the first term as well through V? Please clarify an eventually emphasize this key difference.

→ After the application of the water balance equation at each time step, the random effect is now considered by the variables "month" and "day of the week" for the simulation of turbined water, (which is the used within the water balance equation).

P10 L227. I'm not familiar with this procedure, but I'm assuming that you ends with 58 calibrations. which one is used at the end, or how are them combined? Please clarify.

→ We modified the description shifting from monthly to daily time step (from 59 to 1411 repetitions) referring to the calculation of a mean value of performance metrics (R2 and RMSE) for predicted water turbined.

P11 L242. Why this asymmetric choice instead of 20 and 80? This needs to be justified.

→ We modified the thresholds accounting for the 10th, 20th and 80th, 90th percentiles in order to provide a symmetric and wider description of critical conditions.

P11 L244. Which baseline period? It may worth to specify here.

→ We reported the baseline period for stored water in brackets.

P11 L247. It is not clear to me where the Monte Carlo approach is used. Is it used only for the percentile analysis?

→ We expanded and modified the description of the Monte Carlo approach into the dedicated section "Future critical conditions characterization" providing specific information on its application for analyzing metrics below or above the selected percentiles (lines 396-410 in the marked-up manuscript).

P11 L254. This sentence is a little confusing. I'm assuming that the storage for "flood prevention" is higher than the maximum due to the additional volume stored to prevent a potential downstream flood, but the way the sentence is presented is confusing. Please reword.

→ We reworded the sentence in order to consider the maximum storage including the additional volume for emergency flood prevention (159.30 Mm3).

P12 L257. Figure 4. Please clarify the meaning of the dotted lines (percentiles?). Same in Fig. 5.

→ According to the selection of symmetrical percentiles we modified the dotted lines into colored bands outlining the areas of (baseline) values lower than the 10th, 20th and greater than the 80th and 90th percentiles reporting the explanation within figures caption.

P12 L265. If I understand correctly, this means that the most important component (inflow) is the one that is not affected by the stochastic modeling. This needs to be further discussed and emphasized in order to clarify the value of using the stochastic approach in similar conditions.

→ We added more information on the inflow from GeoTransf in Section 2.1 (lines 239-251 in the marked-up manuscript). We also moved and rephrased the sentence pointed in this comment in the first part of the discussion, and we referred to it within the limitations of this study (Section 4.1) related to the GeoTransf application.

P13 L270. In the whole section, statistical significance of the changes need to be reported. Even changes with different sign may not be that different if the changes are not statistically significant (also accounting for natural fluctuation within the 30-year window). See Welch test or similar.

→ We introduced statistical tests between baseline and future values integrating it in the text of section 2.4 and reporting a summary table for both the whole time series (hence referring to Figure 6) and on monthly averages values (hence referring to Figure 7) in section 4.2 for future projections.

P 14. Figure 6. I'm not sure that showing each single year is relevant, since these are projections. I would find a way to show time slices rather than annual/sub-annual values.

→ We replaced this figure providing boxplots for each 5-years' time slices.

P15 L302. It would be interesting to integrate the analysis with the result in terms of contribute to the annual storage. Differences in rainy months may be much more relevant for the total storage that during the dry months, even if the percentage differences are comparable.

→ Thanks for bringing this in the discussion as in our initial discussions we previously computed original Fig 7 and 9 rescaling them to the total annual water turbined and total annual storage (hence providing information on average month effect over the whole years). However, since these results have a similar trend to the monthly percentage variation figure (Fig 7 and 9), we opted to only provide this latter.

P16 L312. Number of events is not the only relevant metric. For drought/water scarcity: are those events longer? what about the number of deficit days per year? Is the total/average deficit of these events increasing? For flood: is the max surplus increasing? Those are examples of simple analyses that can be added without much effort.

→ We introduced other metrics to characterize critical conditions of low and high volume stored (adapted from Vogt et al.2018) reporting them in a dedicated section in the methodology (section 2.5) as well as for the results (section 3.3) providing information on the metrics used:
- absolute and relative frequency (changes in overall number of months below/above different thresholds and changes in the number of months below/above different thresholds over 14 years' time-window),
- duration (maximum number of consecutive months below/above the selected thresholds) and
- relative severity (average annual volume below/above the selected thresholds over the total volume stored over the 14 years' time-window) both for the baseline and future projections.

P16 L322. You should add also the same statistics for the reference period. How those number differ from the reference values? Are the differences in the short term (2021-2050) in line with what is already observed? Is the current condition (2021) very different from the reference period? Some insights on that would be useful to understand the reliability of these projections.

→ We calculated the defined metrics for the baseline adding them to the results (section 3.2 in the revised document and 3.3 with reference to the Figure 8 and 9) and discussion sections for a more robust interpretation of future conditions.

P17. L340. Please check the use of the term "negatively" here (and in the rest of the discussion as well). This can be interpreted as either mathematically (sign minus) or qualitatively (to worse). I suggest to rephrase.

→ Thanks for pointing that out and we modified the whole sentence (lines 631-634 in the marked-up manuscript).

P18 L353-354. What about flood? Can increasing storage during winter be a problem in case of flood? Please elaborate.

→ Thanks for pointing out such an interesting side-effect due to the implementation of adaptation strategies such as an earlier water accumulation. We clarified how such strategy can affect flood events (lines 654-658). In particular, we referred to:

  o The results on the number and magnitude of months with positive variations with respect to negative ones which are limited in number for all scenarios.

  o The additional storage for "flood prevention", which availability needs to be always ensured and managed together with the Civil Protection to prevent a potential downstream flood, even in case strategies of earlier reservoir accumulation would be considered.

P18 L356. Increasing in high and low volumes? I'm not sure that this sentence is in line with your results. Please clarify.

→ We modified the sentence in order to consistently refer to the obtained results.

P18 L359. I think that the Monte Carlo results may give an idea on the robustness of such changes, but this is not discussed at all at the moment.

→ Thanks and we elaborated and expanded the use of the Monte Carlo approach into the discussion (lines 670-675 in the marked-up manuscript).

P18 L364-365. This is more a conclusion that a discussion of the results.

→ We moved the sentence in the last part of the conclusions (lines 753-755 in the marked-up manuscript).

P18 L375. This is a very important point that needs to be highlighted in the methodology as well.

→ Thanks and we reported this information in Section 2.4 (Future projections and statistical analysis, lines 364-367 in the marked-up manuscript) to help readers to better understand the conservative assumption on upstream withdrawals.

**Reply to referee 2**

**GENERAL COMMENTS**

A. The authors state that an innovative stochastic System Dynamics Modelling (SDM) approach is applied to simulate water stored and turbinated in the S. Giustina reservoir under current/past and projected climate conditions (p 1, l 18-20). SDM is described as "an approach used in the field of complex system behaviour" useful for analysing large systems and interactions between subsystems (p 3, l 69-79). In presenting the case study, emphasis is put on the importance of achieving "a better understanding of the complex interactions in the S.Giustina water management", and it is stated that "SDM can help to depict the S.Giustina reservoir dynamics and its responses to future pressures, including climate change and anthropogenic factors" (p 5, l 127-132). However, the SDM approach simulates reservoir volume and outflow using reservoir inflow produced by the GeoTransf hydrologic model as input (Fig. 3 box 3, p 8, l 168, and l 181-184). Its implementation consists of two regression models simulating reservoir volume as a function of inflow and month (Eq. 1) and outflow as a function of inflow, past volume and month (Eq. 2). This simple approach seems not to correspond to the provided SDM definitions.

→ Thanks and we modified some of the initial expectations (lines 65-67 and 102-106 in the marked-up manuscript) while referring to novel SDM applications and its possible modular integration in a wider modelling context leading to (more) complex systems analysis (Duran-Encalada et al., 2017; Gohari et al., 2017; Sušnik et al., 2018, Malard et al., 2017; Stave, 2010; Davies and Simonovic, 2011).

B. The authors argue that the novelty of their approach lies in the application of SDM in a probabilistic/stochastic framework, in order to account for the uncertainties of future climate scenarios (p 2, l 51-57; p 4, l 94-95; and p 8, l 174-176). However, the stochastic (Monte Carlo) approach is given a very short and unclear description at the end of Sect. 3 "Material and methods" (see specific comment 21). Moreover in the results section, the outcomes of the stochastic sampling shown in Fig. 6 and 8 are either not discussed or only mentioned briefly (see general comments G and H).

→ Thank you for pointing that out. We expanded the description of the MonteCarlo approach in Section 2.5, the results coming from its application in section 3.3 and the final discussion in section 4.

C. The role of anthropogenic pressures is mentioned in several parts of the manuscript (e.g. p 2, l 61; p 4-5, l 108-116). The use of water demand/withdrawal data is mentioned briefly only on p 11, l 235 ("GeoTransf simulations considered unchanged maximum water withdrawals in the Noce catchment") and on p 18, l 374 (similar statement). However, these data are not described nor referenced in the materials and methods section. Most importantly, their role in the modelling framework is not described at all. The conclusions read that "Future model expansions include water demand from multiple human activities" (p 19, l 401-404). The authors should clarify whether non-hydropower water abstractions are simulated and, if applicable, describe how these are included in the modelling framework.

→ Thanks for your comment. We referenced to the conservative assumption of non-hydropower water abstractions upstream of the reservoir considered at their maximum value (conservative assumption in

GeoTransf) in section 2.4, while providing a wider context on the different sectorial withdrawals in the case study description (section 1).

D. SDM components (stocks, flows, converters, connectors) are introduced in Sect. 1. However, the terms "stock", "converter" and "connector" do not appear in the rest of the manuscript. These terms should be used when describing the methodology (throughout Sect. 3 and in particular in Fig. 2) to clarify why SDM is relevant for this study. Moreover, the definitions and explanatory examples of "converters" and "connectors" are unclear (p 3, l 71-73). Evaporation rates, which are used as examples of "converters", may also fulfil the definition of "flows" ("variable's rate of change"). Instead, should the parameters of evaporation equations be regarded as "converters"? Otherwise, are "flows" fluxes of material between "stocks" simulated by the system (e.g. reservoirs and rivers) rather than fluxes leaving the system (e.g. evapotranspiration)? The example of "linking of temperature variations to evaporation rate" may indicate that "connectors" act in the opposite direction of "converters", but it is not clear how these two categories are qualitatively different. These definitions should be clarified, especially because the book by Sterman et al. (2000) is not an open-access publication.

→ We added reference to the SDM terms in figure 3 as part of the legend and in the caption explicitly referring to the single variables. The explanatory example for converters was changed in: "converters - parameters influencing the flow rates (e.g. temperature variable acting to alter evaporation from a water body) and (iv) connectors – as arrows transferring information within the model (e.g. linking the monthly effects on reservoirs' water discharges) (Sterman et al., 2000)." (lines 68-74 in the marked-up manuscript) We trust the new text is clearer in its meaning of the terms, offering better distinction between flows and converters.

E. The description of the "causal loop diagram" (Sect. 3.1) is too brief and needs to be rewritten as the roles of several elements described in the text and shown in Fig. 2 are not understandable. In general, the authors should clarify to which extent the arrows in the diagram correspond to the flow of information between models shown in Fig. 3: if these representations overlap, then Sect. 3.1 and 3.2. could be merged into a single section. More specifically, in what ways do "soil" and "land-use" represent system vulnerability? Also, it is not clear how the "month of the year", "water turbined" and "reservoir volume" represent exposure. Finally, the "critical states" are undefined.

→ Following both referees comments and the different overalps with figure2, we decided to remove Figure 2 and to give more space to following methodology sections (e.g. water balance equation application, statistical test implementation, different metrices of evaluating low and high volume stored).

F. In Table 3 (Results), average outflows are between 8% and 9% smaller than average inflows. These discrepancies, which are not discussed in the manuscript, may imply (i) that the closure of the reservoir water balance is not guaranteed, as over long time periods total inflow and outflow should be very close (except a change in storage that would be very small compared to the cumulated flows); or (ii) that the difference between inflow and outflow is lost via evaporation, percolation, non-turbinated releases, or abstracted for other uses than hydropower. However, none of the latter losses/uses are mentioned in the manuscript. The combined use of Eq. 1 and 2 (or the corresponding models #3 and #4 in Table 2) to

model volume V and outflow Qout may violate the reservoir water balance: once Qout(t) is calculated with Eq. 1, as V(t-1) and inflow Qin(t) are also known from previous calculation (or initialisation) and input data, then V(t) should be obtained by applying the reservoir mass balance: V(t) = V(t-1) + Qin(t) - Qout(t), where V(t) is the volume at the end of month t, and Qin(t) and Qout(t) are average inflow and outflow during month t. The authors should clarify how the closure of the reservoir water balance is guaranteed, as this is a prerequisite for a reservoir simulation model: to do so, they may replace the expressions in Table 2 with understandable mathematical equations, and define the "f" functions in Eq. 1 and 2 (see specific comments 17-19). If the water balance closure is not guaranteed by this approach, the reservoir simulation model should be revised.

→ We modified the modelling approach to iteratively integrated the water balance equation at a daily resolution with operational rules for minimum ecological flow and emergency releases. Values were then averaged at monthly resolution to better provide support for large reservoir management in long-term conditions with climate change effects.

G. The presentation of results needs to be improved. For example, Fig. 6 is not discussed in the text: it is referenced only once on p 13, l 272 while discussing the average values presented in Table 3. Figure 6 contains 12 time series plots and seems to use the grey shaded area to show stochasticity, which is presented as a major component of the methodology. Therefore, Fig. 6 should be discussed extensively, focusing in particular on the value of the information provided by the stochastic approach for the water scarcity risk assessment. Moreover, the confidence interval is only mentioned in the caption of Fig. 6 without specifying the confidence level nor the assumptions underlying its calculation, thus hindering the interpretation of the plots. Also, the apparently smoothed lines in volume and outflow plots (Fig. 6) are not defined nor discussed in any part of the manuscript. For other examples, see specific comments 22-27.

→ Thanks for the feedback. We better described the figures in section 3.2 covering the reported information. In agreement with the comment from Reviewer #1, we integrate Figure 5 with better caption information and modify the figure in order to replace single projected values with aggregated 5-year time slices.

H. The emphasis on the probabilistic/stochastic framework is not followed by a thorough discussion of the uncertainty intervals derived from the Monte Carlo sampling. For example, Fig. 7 and 9 only show average variations. Moreover, although some outcomes of the stochastic approach are shown in Fig. 6 (grey areas) and 8 (box-plots), these are either not discussed at all (Fig. 6) or mentioned only briefly (p 16, l 310-315) in the manuscript.

→ We integrated results (section 3.3) and discussion (Section 4) on the MonteCarlo sampling approach and its application with reference to Figure 8 and 9 as this was also pointed out by Reviewer #1.

**SPECIFIC COMMENTS**

1. p 1, l 20: Based on the information provided in Sect. 3.2, the sentence "The integration of outputs from climate change simulations as well as from a hydrological model and statistical models into the

SDM is a quick and effective tool to simulate past and future water availability and demand conditions." should be replaced by a more informative statement on how the components of the model tool-chain are assembled, e.g. explaining briefly the input/output flow between model components.

→ Thanks and we rephrased the sentence clarifying the use of a model chain (lines 20-24 in the marked-up manuscript

2. p 1, l 21-23: Add simulation periods to "short-term", "long-term" and "baseline".

→ Thanks for pointing that out. We agree and will add them (lines 25-28 in the marked-up manuscript).

3. p 1, l 25 and throughout the text: Replace "quantiles" with "percentiles" when using values within 1 and 100.

→ Thanks and we modified them throughout the text.

4. p 1, l 27-29: This statement seems not to be supported by the results, as no other water-demanding sector than hydropower is considered in this study. Moreover, the impact on hydropower production is not quantified, as model output includes reservoir storage volume and outflow but not energy production.

→ Thanks for this comment. We rephrased the statement to provide insights on possible future upscaling and application of SDM to include other water demanding sectors and further characterize water scarcity conditions (lines 43-46 in the marked-up manuscript).

5. p 3, l 89: If possible a scientific publication should be referenced when asserting the "reduced accuracy in comparison with dedicated physically based models".

→ We opted to remove this sentence as it pointed to either consider SDM or physically-based models, while SDMs are useful and often used to explore and connect disparate fields and methodologies.

6. p 4, l 101-105: As detailed data regarding reservoir storage and environmental flow requirements are provided later (p 5, l 126), adding the following information about the Noce river would help understanding the context: catchment area, average discharge, and sectoral or aggregate water demand/abstraction volumes. In particular, water demand/abstraction information would be of crucial importance in light of the mentioned water scarcity events/issues (p 5, l 108-116) and because the S. Giustina reservoir provides water for irrigation and other downstream uses (p 5, l 122-124).

→ Thanks, we added information in section 1 on the catchment area (1367 km$^2$) and the aggregated sectorial demand for the whole catchment as reported in the following table (Percentages subdivision of the licensed water withdrawals per sector (year 2017) where large hydropower plants

(with average nominal capacity greater than 3 MW) water withdrawals are excluded (Data source: Provincial Agency for Water and Energy)):

| Sector | Percentage [%] |
|---|---|
| Agriculture | 16.39 |
| Other | 0.15 |
| Domestic | 3.98 |
| Hydropower | 77.81 |
| Industrial | 0.49 |
| Snowmaking | 0.28 |
| Fish farming | 0.90 |

The high percentage value of 77.81% of licensed withdrawals for hydropower only accounts for the small run-of-the-river plants, hence providing information on the very high use of water for hydropower purposes within the Noce catchment where the S.Giustina dam is the largest reservoir.

7. p 5, l 124-126: The statement on stakeholder concerns about "unused" water releases should be supported with some evidence. Otherwise it should be removed.

→ Thank you and we decided to remove such statement to keep a more neutral position.

8. Figure 2: Clarify whether precipitation is partitioned into snowfall and rainfall as a function of temperature. If so, then rainfall should be added to the diagram. Consider whether the distinction between "Climate-related hazards" and "Hydrological-related hazards" is necessary. The caption seems to be truncated.

→ Following the general comment E, we decided to remove Figure 2 due to its overlaps with Figure 3 and to give more space to the changes in the methodology section (e.g. water balance equation, different metrices of low and high volume stored).

9. Figure 3, legend: The terms "external component" and "internal component" are not discussed in the text: explain with respect to which other elements they are to be considered external/internal. Also, clarify the meaning of "Field/sector" and whether its graphical identifying feature is the dashed line or the fill colour.

→ We modified the text to explicitly refer to the external and internal components in section 2.1 (lines 235 and 239-240 in the marked-up manuscript) and changed the word "Field/sector" into "Modules" both in text and figures to simplify.

10. Figure 3 and p 7, l 160-161: Provide the source of weather station data. Also, describe briefly the bias correction methodology, as "Quantile mapping" is mentioned in Fig. 3 but not in the text.

→ We modified the text and added references to considered E-OBS data and the bias-correction methodology, explicitly referring to be an external component (lines 229-238 in the marked-up manuscript).

11. p 7, 8 and Fig. 3: The baseline simulation period needs to be clarified. From Fig. 3 and p 7, l 160-161, it seems that weather station data (period 1971-2005) are used to bias-correct 1975-2005 COSMO-CLM precipitation and temperature (l 177), which are in turn used as GeoTransf input. However, Table 1 reports that "Geotransf_inflows" cover the period 1981-2010, while in Fig. 3 the hydrological model baseline period is 1980-2010 and the SDM baseline periods are 1999-2004 and 2009-2017. A more detailed description of the input/output flow between model components could clarify these apparent inconsistencies.

→ We corrected the time-window for the baseline in Fig. 3, Table 1 and in the text (lines 234-259 in the marked-up manuscript) in order to explicitly refer to the constrained in data availability for the SDM: the regression model and the water balance application considered the 1999-2004 and 2009-2016 baseline period due to the constrained data availability of the reservoir volume. For this reason, the baseline simulations of water turbined and volume stored considered the observed inflows and not the GeoTransf values.

12. p 8, l 165-180: To understand the application of the GeoTransf model, further details are needed about the input data. In Fig. 3, glacier extension, land use and soil data seem to be model input, but they are either not mentioned or not explained in the text. Add the sources of these datasets. Clarify in particular whether glacier extension is an input or an output variable, as the following sentence seems to imply that glacier state is a model output: "GeoTransf provides a description of the hydrological dynamics within the Noce alpine river catchment, assessing variations in water contributions coming from climate change effects in terms of temperature, soil moisture, glaciers, snow and rainfall" (p 8, l 170-172).

→ We modified and integrated the GeoTransf description with further information and link to the datasets of input data and reference to the GeoTransf application, since it belongs to the external component (lines 239-251 in the marked-up manuscript).

13. p 7, 8: Add the temporal resolution(s) of input data and model output.

→ We added reference to temporal resolution throughout the text in section 2.1.

14. p 8, l 166-168: Clarify what is meant by "these blocks" (temperature and precipitation data? GeoTransf output?) and provide references in addition to the OrientGate project website. Check if the latter should be updated to http://m.orientgateproject.org/.

→We replaced "These blocks'' with "these boxes" in order to create consistency with the used terms within the text, hence referring to box 1 (climate projection) and box 2 (hydrological model) (line 240). We replaced the weblink with the suggested one (line 241 in the marked-up manuscript).

15. p 8, l 182-184: Clarify this sentence. The meaning of "covers" is unclear: are reservoir volume and outflow outputs of SDM? What are the "critical conditions" to which volume and outflow are exposed?

→ We modified the sentence to make it clearer (lines 264-267 in the marked-up manuscript) and removed references to critical conditions as these are specifically addressed in the dedicated section 2.5.

16. p 9, l 187-189: The following input variables are mentioned here for the first time: "hydroelectric energy market price", "water outflows from an upstream dam reservoir". They should be introduced in Sect. 3.2 ("System dynamics modelling set-up and input data") and integrated into the Fig. 3 flow chart. Now they are mentioned briefly in the Supplement without any further detail than units, temporal coverage and source (missing for the upstream dam reservoir outflow).

→ We reported additional information on these variables within the text of section 2.1 ("System Dynamics modelling set-up and input data", lines 252-255 in the marked-up manuscript) and in lines 280-287. In order to simplify and ease readers interpretation Figure 3 only reports the finally selected variables consistently with Table 1. Further information on the variables excluded are reported in the Supplementary material.

17. Table 2: Replace the formulas with mathematical expressions. The current expressions are not understandable. The reader should not be required to learn the R syntax in order to understand what seems to be regression models. This notation may lead to ambiguous interpretations of, for instance, the "lag" operator (how long is the lag?), the "1|month" expression, the "s" function, and so on. Moreover, model parameters are visible in these formulas. The same applies for Table S2 and S3 in the Supplement.

→ We intended to provide readers with insights into the components we included in our assessment showing the R syntax in the caption of Table 2 to enhance transparency and to allow the reader to easily reproduce the general workflow within the R environment. While for the linear models (multi-linear and linear mixed effect model) it would be feasible to report models' formula (see below), gams are a sort of regression model between statistical models and machine learning. Since gams are a semi-parametric model where the degree of smoothness of the single model terms is estimated as a part of fitting (associated smoothing functions automatically fitted using internal cross validation (Wood and Scheipl, 2020; Wood, 2017; Zuur et al., 2009) ), they do not have an explicit form that can be written with math formulae. In other words, gams smooth over the data using kernels that determine the wiggliness of the fitted model. In analogy to machine learning, we think providing information on the many estimated

model components might not add any added value while adding complexity to the manuscript. However, in order to enhance transparency and to allow the reader to easily reproduce the general workflow within the R environment we reported the meaning of the single expression for the lag operator, random effects (1|weekday), (1|month) and the "s" function in Table 2 caption and in lines 293- 299 in the marked-up manuscript.

Here reported the formula for the two considered linear models types.

**Multi-linear model:**

$$y = \beta_0 + \beta_1 x_1 + \ldots + \beta_k x_k + \epsilon$$

With $y$ the response variable, $k$ predictor variables $x_1, x_2, \ldots, x_k$ and $\beta_1, \beta_2, \ldots, \beta_k$ fixed-effect regressors coefficient and $\epsilon$ the residual terms of the model.

**Linear mixed-effect model:**

$$y_i = \beta_0 + \beta_1 x_{1i} + \ldots + \beta_k x_{ki} + b_{i1} z_{1i} + \ldots b_{iq} z_{qi} + \epsilon_i$$

With $y_i$ the response variable in the $i$th group, $k$ predictor variables $x_1, x_2, \ldots, x_k$ and $\beta_1, \beta_2, \ldots, \beta_k$ fixed-effect regressors coefficient, $b_{i1}, b_{i2}, \ldots, b_{iq}$ the random-effect coefficient for group $i$, $z_1, z_2 \ldots z_q$ the random-effect regressors and $\epsilon_i$ is the residual terms of the model for group $i$.

18. Section 3.3.1: Equation (1) seems to suggest that the reservoir volume at monthly time step t is function of inflow at t and month of the year. Why is not V(t) function of V(t-1) as well? This may seem an incorrect representation of the reservoir water balance and should be clarified. Function "f" is not defined, which makes Equation (1) not fully understandable. Also, in the following sentence the expression "grouping effect" should be clarified: "As a random effect, the month of the year was selected (month) for its grouping effect on the recurrent water volume variations on a monthly scale".

→ According to the response to the general comment F, we applied the water mass balance equation (currently equation 1, line 319) at each time step to replicate historical water stored in the reservoir based on simulated turbined water through a linear mixed-effect model. We also modified the explanation of the random effect into "As a random effect, day of the week and month of the year were selected to account for the recurrent water volume variations occurring according to the day of the week and month" (lines 296-299 in the marked-up manuscript).

19. Section 3.3.2: Function "f" is undefined, making Eq. (2) not understandable. Moreover, it seems that a flow (Qin) and a volume (V) are summed together, thus potentially violating the dimensional consistency of the equation.

→ We modified the description of hydropower outflows simulations (please, also refer to reply to question 17) clarifying the meaning of grouping effect in the regressions (lines 296-299 in the marked-up manuscript) and referring to different references in case of information on the mathematical background of the statistical techniques.

20. Section 3.4: The authors need to state explicitly which model parameters are calibrated. The description of the "forward time-window approach" needs to be improved. It seems that the whole procedure consists of a sequence of 59 successive model calibrations, each based on an increasingly longer training dataset. However, it is unclear how information from iteration n-1 is used in iteration n. Also, what algorithm is used to search the parameter space?

→ We explicitly referred to the statistical model for "turbined water" (line 344 in the marked-up manuscript) in current section 2.3 to clearly point to a mean RMSE value (lines 353-356 in the marked-up manuscript) coming from all repetitions as a more robust estimation of model performance through multiple temporally independent subsets compared to common one-fold non-temporal validation methods. Differently from physically-based models, there is no a priori information on the parameter space definition since this is directly derived from the available input data parameter range.

21. p 11, l 246-249: The description of the Monte Carlo approach needs to be clarified. The authors should clarify how the sampled volume time series are built, explaining in particular how the relevant characteristics of simulated volume are preserved (e.g. autocorrelation and seasonality).

→We clarified the description of the MonteCarlo approach providing more information on the sampling steps (time series of continuous 14 years as for the forward time-window approach) (section 2.5, lines 396-410 in the marked-up manuscript). Autocorrelation and seasonality are preserved as subset replications are extracted from the simulated volume values and moving progressively through the whole 30-years period.

22. p 11, l 255 and Fig. 4: Correcting simulated volume values to the maximum storage hides model overestimations. Moreover, the time steps of these corrections are not shown in Fig. 4, thus not allowing the reader to distinguish between simulated and corrected values. For transparency and to allow the evaluation of the modelling approach, volume values greater than the maximum allowed for flood should be shown in Fig. 4.

→ The implementation of the water balance equation at daily temporal resolution allowed to implement a better description of the different outflows values (e.g. emergency release and minimum ecological flow) that regulate the stored volume (described in section 2.2.2, lines 319-325 in the marked-up manuscript). For this reason, simulated volume values are not corrected anymore.

23. p 12, l 263-266: The sentence "[...] low values of reservoir volume compared to the monthly inflow values" seems incorrect: the active storage volume is 152.4 Mm3 (p 5, l 121) and the average baseline monthly inflow is 71.38 Mm3/month (Table 3). Moreover, if storage were small compared to monthly inflow, then the analysis should be carried out using a sub-monthly time scale (e.g. daily), as the reservoir would not be able to significantly regulate the flow at monthly scale.

→ We modified the modelling accounting for daily resolution in the water balance equation and provided information on the suitability of using a monthly time resolution for long-term trend

analysis in large reservoir management (Solander et al., 2016; lines 416-418 in the marked-up manuscript).

24. Figure 5: What are the horizontal dashed lines?

→ Horizontal dashed lines were substituted by colored bands for the selected percentiles and their meaning integrated in figure 3 and 4 captions and in the text (lines 420-421 in the marked-up manuscript).

25. p 13, l 273-275: 2021-2050 RCP8.5 average inflow is 7.5% smaller than the baseline, while precipitation is 1.4% larger (Table 3). Is this due to increased evapotranspiration caused by the relatively large temperature increase? The authors should discuss the impact of increasing temperatures on local hydrology.

→ Thanks and we referred to evapotranspiration increases within the text (lines 454-456 in the marked-up manuscript).

26. Table 3: the "Δ [%]" values seem wrong in several cases, e.g. the relative difference between 2021-2050 RCP4.5 and baseline temperature is 27.6% and not 0.5%. Similar errors occur for all other temperature changes. See also p 15, l 289.

→ We corrected the values for temperatures while modified those for outflows and volume according to the modified modelling results.

27. p 17-18, l 344-346: Define "slow onset conditions of water availability" in the context of the presented results.

→ We rephrased the sentence moving it at the beginning of the discussion and pointing to "conditions of reduced water availability" in order to provide a clearer terminology (lines 621-623 in the marked-up manuscript).

28. p 18, l 356-357: The sentence "At the same time, high volume events decrease [..]" seems to be redundant with the previous one, i.e. "[...] increasing number of future water scarcity conditions of high and low volumes stored".

→ We modified the whole paragraph deleting the sentence mentioned in the comment while referring to the new results.

29. p 18, l 369: The sentence "Accounting for the GeoTransf application means relying on a very accurate water evaluation within the catchment" should be supported with some evidence/references. Moreover, the expression "water evaluation" seems ambiguous: the authors should mention what specific characteristics of the water system are reproduced accurately (e.g. river discharge?).

→ We added reference to the GeoTransf model and edit the sentence poiting to water streamflows (lines 689-690 in the marked-up manuscript).

30. p 18, l 376-377: Explain how the precipitation data used to force the GeoTransf model may miss intense precipitation episodes. This is not clear from the presentations of input data and results.

→ The sentence in l 376-377 aims to highlight the focus of this study on trend variations at monthly scale on short- and long-term which can affect the S.Giustina water turbined and volume stored (e.g. Fig 7 and 9). We modified the sentence (lines 700-703 in the marked-up manuscript) to explicitly refer to the long-term trends of volume conditions consistently with the objective of this study.

31. p 19, l 378: It seems that the authors refer to the reservoir volume time series length (14 years) as a factor that limits the model predictive performance. Model performance was evaluated using RMSE and R2 (Sect. 4.1). However, it is not clear how time series length negatively affects the goodness-of-fit of the regression models (Table 2). In contrast, small training samples may lead to over-fitting, which then can cause poor model ability to simulate time periods over which the parameters were not calibrated.

→ We wanted to point at implementation of advanced techniques (i.e. moving window) to better estimate the models performances in case of limited data availability (volume data compared some of the other variables, e.g. inflow). Nevertheless, we have understood this sentence can be misleading in the limitation section and we deleted it in order to simplify and help the readers.

32. p 19, l 385-387: Was the monthly time resolution chosen because of the lack of sub-monthly (e.g. daily) data or following other considerations (as it seems from the brief mention on p 9, l 193-195)? This choice should be discussed in detail in the materials and methods section as it has a significant impact on the modelling framework, e.g. the authors have not used the energy price as a regressor when modelling outflow.

→ We modified the modelling accounting for daily resolution in the water balance equation to better capture volume regulations (lines 264-267 in the marked-up manuscript). We then aggregated these results to a monthly resolution being a suitable time resolution for supporting reservoir volume management over long periods considering climate change effects (Solander et al., 2016).

**Reply to Mario Wetzel**

1. Figure 2: The CLD in figure 2 shows in a very comprehensive way the interlinkages of different factors that finally lead to the specific risk. However, the figure does not contain feedback loops. Could you please clarify the difference here to an Impact Chain (cause-and-effect chain) or a Directed Acyclic Graph such as used in Bayesian Network modelling. Did you consider and model direct feedback loops in your study?

The causal loop diagram (CLD) was developed as an initial conceptual framework and feedback loops were not identified in this specific case. It has indeed common points to the so-called Impact Chains (e.g. reference to the IPCC AR5 risk components) or to the Bayesian Networks (e.g. focus on variable's probabilistic representations) although system dynamics modelling aimed to characterize the time-dependent functions affecting the variables of the system under study.

2. P. 2, 54ff: The manuscript mentions the benefits of the methodology to better understand the dynamics between anthropogenic and environmental processes. Looking at figure 2, I associate land-use, water turbined and reservoir volume to factors that are driven by anthropogenic processes and the interplay of anthropogenic and environmental processes. However, pertaining to exposure, "water turbines" and "reservoir volume" are solely driven by "month of the year". "Month of the year" remains a black box, in a sense that drivers that specify the differentiated role of intra-annual dynamics are not further specified. Is "month of the year" solely driven by precipitation variations, or also informed by differentiated economic activities, water consumption etc.?

Thank you and after internal discussion on referee's comments we decided to delete figure 2 and give more room to figure 3 in order to better explain the model chain and the underlying variables. Nevertheless, "water turbines" and "reservoir volumes" represent the elements exposed to potential risk conditions, driven by reservoir management during different months of the year as well as day of the week (considered as random effect in the linear-mixed effect model).

3. P. 19, 383ff: Can the application of scenarios regarding the reservoir management be integrated here, or an alternative assumption (compared to the stationary) based on potential societal changes?

Thanks for this comment. The statistical approaches allowed to replicate past turbined outflows and the consequences on the stored volume. Such rule was kept unchanged in future scenarios for investigating potential critical states conditions of water turbined and reservoir volume. However, we see the possibility to play with future conditions of water inflow to the reservoir according to different scenarios of upstream anthropogenic water withdrawals in future works since the upstream-downstream water management is an important feature to potential future disputes.

**References:**

Bellin, A., Majone, B., Cainelli, O., Alberici, D., Villa, F., 2016. A continuous coupled hydrological and water resources management model. Environ. Model. Softw. 75, 176–192. https://doi.org/10.1016/j.envsoft.2015.10.013

Duran-Encalada, J.A., Paucar-Caceres, A., Bandala, E.R., Wright, G.H., 2017. The impact of global climate change on water quantity and quality: A system dynamics approach to the US-Mexican transborder region. Eur. J. Oper. Res. 256, 567–581. https://doi.org/10.1016/j.ejor.2016.06.016

Gohari, A., Mirchi, A., Madani, K.D.E. of C.C.A.S. for W.R.M. in C.I., 2017. System Dynamics Evaluation of Climate Change Adaptation Strategies for Water Resources Management in

Central Iran. Water Resour. Manag. 31, 1413–1434. https://doi.org/10.1007/s11269-017-1575-z

Huang M., Elshorbagy A., Barbour SL., Zettle J., Si BC. 2011. System dynamics modeling of infiltration and drainage in layered coarse soil. Canadian Journal of Soil Science. 91: 185-197. DOI: 10.4141/cjss10009

Malard, J.J., Inam, A., Hassanzadeh, E., Adamowski, J., Tuy, H.A., Melgar-Quiñonez, H., 2017. Development of a software tool for rapid, reproducible, and stakeholder-friendly dynamic coupling of system dynamics and physically-based models. Environ. Model. Softw. 96, 410–420. https://doi.org/10.1016/j.envsoft.2017.06.053

Solander, K. C., Reager, J. T., Thomas, B. F., David, C. H. and Famiglietti, J. S.: Simulating human water regulation: The development of an optimal complexity, climate-adaptive reservoir management model for an LSM, J. Hydrometeorol., 17(3), 725–744, doi:10.1175/JHM-D-15-0056.1, 2016.

Stave, K., 2010. Participatory system dynamics modeling for sustainable environmental management: Observations from four cases. Sustainability 2, 2762–2784. https://doi.org/10.3390/su2092762

Sterman, J.D., Burr Ridge, B., Dubuque, I., Madison, I., New York San Francisco St Louis Bangkok Bogota Caracas Lisbon London Madrid Mexico City Milan New Delhi Seoul Singapore Sydney Taipei Toronto, W., 2000. Business Dynamics Systems Thinking and Modeling for a Complex World.

Sušnik, J., Chew, C., Domingo, X., Mereu, S., Trabucco, A., Evans, B., Vamvakeridou-Lyroudia, L., Savić, D.A., Laspidou, C., Brouwer, F., 2018. Multi-stakeholder development of a serious game to explore the water-energy-food-land-climate nexus: The SIM4NEXUS approach. Water (Switzerland) 10. https://doi.org/10.3390/w10020139

Vogt, J. V., Naumann, G., Masante, D., Spinoni, J., Cammalleri, C., Erian, W., Pischke, F., Pulwarty, R. and Barbosa, P.: Drought Risk Assessment and Management. A conceptual framework., 2018.

Wood, S., Scheipl, F., 2020. gamm4: Generalized Additive Mixed Models Using mgcv and lme4.

Wood, S.N., 2017. Generalized Additive Models : An Introduction with R. Gen. Addit. Model. An Introd. with R, Second Ed. 1–476. https://doi.org/10.1201/9781315370279

Zuur, A.F., Ieno, E.N., J., W.N., Saveliev, A.A., Smith, G.M., 2009. Mixed Effects Models and Extensions in Ecology with R, Springer.

---

## Author Response (AR2)

**Referee 1**

Overall, a better description of the anthropogenic effects, in both the hydrological modeling of the inflow and in the SDM, is needed. This is particular important for future projections, where you need to emphasize how your hypotheses (no changes?) compare to the expected future behavior or potential SSP compatible with RCP4.5 and RCP8.5. These background information would be very useful to interpret the obtained results (i.e. differences between short and long terms and RCP4.5 vs. RCP8.5) and also to improve the discussion section.

Also, as suggested by one of the previous reviewer, and following the revision of the methodology section, I would downplay the use of terms as "innovative", "novel" and similar throughout the text (not limited to the examples reported in the specific comments).

We would like to thank the Referee for their constructive feedback. Please find our responses (in blue) to the main points raised (shown in black) below. Here reported the main improvements:

- We integrated information on the anthropogenic demand and assumption for future simulations
- We change the representation of future values for inflow, outflow and volume into boxplot for 30-years
- We downplayed the text on SDM making it simpler

Minor Comments

L21. please clarify the sentence "creation of statistical models into".
We modified and integrated the text with the outputs and aim for the statistical model.

L21. The study aims at simulating…
Thank you, we have corrected the text accordingly

L22. volume, as well as…
Thank you, we have corrected the text accordingly

L23. "whole 30-years of simulations…". It is not clear which 30-year period are you referring to here. Especially followed by the sentence on the results for 2021-2050 and 2041-2070. Do you mean something like "Average results on 30-year slices"?
We have corrected the text accordingly

L24-27. It is worth to mention here what is the reference (i.e. present) period for these differences. In any case, differences between RCP4.5 short-term and RCP8.5 long-term are very small. Why? If this is discussed in the text, it needs to me briefly mentioned here.
We added reference to the baseline (lines 26-27) and a brief sentence on factor driving the similar trend for RCP4.5 short-term and RCP8.5 long-term (lines 28-29 in the marked-up manuscript). This description was also added in section 3.2 with reference to climate projections from Bucchignani et al., 2016.

L27. January to April.
We have corrected the text accordingly (lines 30-31 in the marked-up manuscript)

L29. Report months in which the mins occur, similarly to the maxs.
We have corrected the text accordingly (line 32-34 in the clean manuscript version)

L30-31. Even if I understand that "increase in terms of low volume" and "decrease of high volume" refer to frequency, duration and severity (i.e. more frequent, longer and severe deficit), the sentence may be easily misunderstood. I suggest to reword.
We have moved this part closer to the initial metrics terms to be sure it is referring to them (lines 34-36 in the marked-up manuscript)

L54. It seems that a word is missing after snowpack (like dynamic or something).
We have added the missing word (line 61 in the marked-up manuscript)

L78. I suggest to reword point ii) to better highlight what is the limitation (since this a what the list is all about).
We have removed the last part of this sentence so to simplify and focus on the mentioned limitation (line 86 in the marked-up manuscript)

L84-87. Since stochastic SDM is not a new approach, I suggest to add few references and examples on how a stochastic component can be added to SDM, similarly to the overview that you did on SDM in general.
Thanks for your comment, we have integrated the text accordingly (lines 93-94 in the marked-up manuscript) adding reference to two papers dealing with this approach (Taylor et al., 2009; Ford, 2005).

L89. I suggest replacing effects with projections.
We have corrected the text accordingly (line 107 in the marked-up manuscript)

L112-119. "Temperature increase, …. (i.e. hydropower plants)." is more suited for the introduction, to explain the reasoning behind the study, than for methodology.
We understand the referee comment and we moved the suggested part into the introduction (lines 99-106).

L129-134. same as above. Not really relevant for this section, but it can be partially used in the introduction to better explain the context of the study.
Thanks for pointing this out, we moved some of this part into the introduction (lines 112-114) and changed it.

L136-137. Some wording here, such as "novel", "robust" and "particularly useful" is excessive. I suggest to keep the text as simple as possible.
We removed those words as suggested to keep the text simple.

L140. Similarly, "innovative" is not really needed here.
We removed it too.

Fig. 2. Clarify in the figure why there are two baselines in the SDM.
We substituted "&" with semi-columns in the baseline and added brackets in Figure 2 pointing to the use of a single baseline having a gap from 2005 to 2008 as added in Figure 2 caption and in line 212-213 in the marked-up version.

L170. Please provide some details on how the anthropogenic effects are modeled in GeoTransf (i.e. water use modeling). Is the basin highly regulated upstream of the reservoir? Is any of these anthropogenic effects accounted in the inflow? Similarly, is any of this modified for the projections?

We added additional information on water withdrawals in GeoTransf (lines 201-202 and in lines 290-291 of section 2.4 in the marked-up manuscript). Withdrawals upstream the reservoir were considered affecting the inflow and set unchanged to the maximum discharge for future simulations given the water infrastructure constrains in order to account for future increases of water demand from domestic and agricultural needs.

Please provide here a brief summary of the validation (i.e. accuracy, bias, other metrics).

We provided the resulting performance metrics (lines 197-198 in the marked-up manuscript) while referring to the main publications for the full explanation of the calibration and validation methods and results (Bellin et al., 2016).

L178. What are the hypotheses in the projections? Is climate the only thing that is changing? Any change in population, agricultural area, water used included? Please clarify (also in the context of expected future evolution of the area), since this is a key factor in interpreting the results.

Yes, future projections considered only climate as changing while the assumption on the maximum water withdrawals for water uses encompasses possible water demand increases due to tourists and agricultural needs. We added this information in lines 205-207 in the marked-up version.

L179. Other variables (drop input).

We have corrected the text accordingly

L181-184. This is not clear to me. Does it means that the SDM was forced (inflow) with a combination of modeled and observed data depending on the period? Please clarify and explicitly say what is used in which period.

Yes, we changed the sentence into: "the SDM was forced with past observations of inflow to S.Giustina (with data available for the period 1981-2016) and with GeoTransf values for future simulations due to the limited temporal overlap of past GeoTransf values (over 1981-2010) with observations of the S.Giustina stored volume (over 1999-2004; 2009-2016, Table 1)" (lines 213-216 in the marked-up version)

L199. Please quantify "low predictive performance" (i.e. xx statistical metric lower than yy).

We modified the sentence providing information while referring to the Supplementary material since the variables selection was not based on a single performance criterion, but it was based on the interpretation of three criteria (i.e. (i) data availability to the maximum target time period, (ii) correlations among the explanatory and response variables and (iii) the selection of the most parsimonious model). (lines 229- 233 in the marked-up version)

L202. The best models for each model types. Pleas reword, maybe something like "the best relationships/regressions".

We have corrected the text as suggested (line 235 in the marked-up version)

L204. I would invert this sentence, to emphasize that it was chosen thanks to being the best at monthly scale, and also the good performance at daily scale. Anyway, I do not want to argue with your choice,

but you should come up with a combined metric that show that model #2 is better than, let say, model #4. Right now this choice seems a little arbitrary.

We have corrected the text as suggested and we understand that presenting both daily and monthly results could confuse readers. For this reason, we decided to keep the monthly results while reporting daily results in brackets within the text and in Table 2 as well as adding this information in Table 2 caption.

L228. Please emphasize that here you are assuming that no change in how ecological flow are treated will occur in the future.

We have corrected the text as suggested (lines 264-266 in the marked-up version).

L250-252. This should be also mentioned above, in the model description. What does it means that this is a conservative assumption? Are population and water use expected to increase or reduce in the future in this area? Do you have any reference to back up this assumption?

Thanks for bringing this in the discussion and we added information in lines 205-207 in the marked-up version. Future water withdrawals were simulated at their maximum possible abstractions given the current infrastructure and water network (Bellin et al., 2016). This can be considered as a conservative assumption to account for potential future increases in water demand which is expected to rise for agriculture, due to the increase in temperature and hence evaporation and evapotranspiration, as well as for the domestic sector (La Jeunesse et al., 2016). Nevertheless, comprehensive analysis of the combined effects of water demands from multiple sectors (i.e. agriculture, domestic, etc) in future projections are currently missing for this area.

Table 3. I appreciate the extension of the analysis to additional metrics. However, what is the reason to include both absolute and relative frequency, if the relative values are derived by dividing for the same fixed values of 14 years everywhere? I would keep only the relative frequency and remove the absolute one. Similarly, you can divide the severity for 14 years and get a average annual severity (not necessary, but more consistent).

Thanks for the comment, we have removed absolute frequency and divided severity by 14 years in Table 3 and throughout the text for keeping a consistent reference to annual metrics as suggested.

L274. Maximum duration, rather than just duration.

Thank you and we have corrected the text accordingly (line 312 in the marked-up version)

L276. This last metric provides information…

We have corrected the text accordingly (lines 314-315 in the marked-up version)

L294-295. …to support reservoir… effects. This sentence is unnecessary, especially since you are discussing only the baseline run here.

We have removed this last part of the sentence accordingly (lines 332-333 in the marked-up version)

Figs. 3 and 4. There seems to be a limited capability to capture low values, such as the one in 2003 (a well known drought year in Europe). This needs to be highlighted and discussed.

We have highlighted and discussed two cases of very high turbined outflows in summer 2001 and very low stored volume in spring 2003 to cover two specific cases where the modelled values did not well represent real values (lines 344-349 in the marked-up version). For the very low volume in spring 2003, the S.Giustina reservoir was emptied due to construction works at the dam, while the 2003 drought

effects are still visible with monthly values of May and July ranking as the second lowest value for each month of these months over the available data.

Fig. 5. I still think that even 5-year slices are too small for such analyses on projections. Even if there are differences between two 5-year periods, what do they say about projections? The two 30-year periods are already overlapping, and the differences are quite small. Just report the same plot for the two 30-year periods (and two scenarios) and you will have a much easier to read figure with much more robust results.

Thank you for the feedback. We modified the figure into 30-year slice as suggested to simplify the representation.

Please clarify that colors are the same as in Fig. 4.

We have added this clarification in Figure 5 caption

L313. Please report just one digit in the percentages (as in the Table). Please apply throughout the text.

We have corrected the text accordingly leaving more digits only for predictive performance indicators (i.e. R2, RMSE and p-value).

L313-315. The way this concept is formulated is confusing. I would say changes in RCP4.5 are rather similar for short and long terms, whereas changes are bigger in the long term for RCP8.5 compared to short term.

We have corrected the sentence following your suggestions (lines 355-356 in the marked-up version), reporting percentage values only for RCP8.5 short and long term and referring to Table 4.

In general, small change in short term for RCP8.5 need to be discussed (just due to increase precipitation?).

Thanks for pointing this out and we have modified and provided information on RCP4.5 and 8.5 values coming from the COSMO-CLM projections referring to the publication of Bucchignani et al., 2016 where total precipitation show a consistent trend with results for the variables inflow, outflow and volume (lines 356-358 in the marked-up version).

L324-332. I'm not sure if looking at trend within the 30-year window is a good practice, since statistics on 5 years are not robust enough for climate projections. As I said above, I will focus on comparing short and long term.

Thanks for pointing this out and we have modified the text according to the new Figure representing boxplots over 30-years time slices (lines 373-382 in the marked-up version).

L360. This should be Figure 7.

We have corrected the text accordingly (line 409 in the marked-up version)

Figs. 8 and 9. remove absolute frequency, since the results are identical to relative frequency (as expected).

Thank you and we removed absolute frequency from Figures 8 and 9.

L384. …longer maximum duration…

Thank you and we have corrected the text accordingly (line 433 in the marked-up version)

L392. last longer in the case of the most extreme events (you are still looking at maximum duration only). Please carefully check the whole section for similar inconsistencies.

Thank you for the feedback and we modified this sentence to explicitly refer to the most extreme events as suggested and also modified lines 441, 455 and 508-509 in the marked-up version for consistency.

L426. I would replace trend with behavior/dynamic, since you are not really looking at trends here. Also, an adjective is needed in this sentence in my opinion (represents well/perfectly, how?).
We have corrected the text accordingly and moved it to the conclusions (line 550 in the marked-up version).

L426-433. This may fit better as the start of a conclusion section than of a discussion.
We moved this part at the beginning of the conclusion (line 550-554 in the marked-up version).

L449. It would be nice to simulate this, by changing the way water usage are modeled in the future, and having a quantification of the actual effect on summer deficits.
We agree on the importance of testing adaptation strategies to quantify their effects, but we believe this addition would change the current focus/objective of the manuscript.

L475. The climate model…. Are your referring to this specific climate model (COSMO-CLM)? Please clarify. In any case, the use of multiple models is important to account for uncertainty in the future, since even if a climate model reproduces well the climate of the past, this does not guaranty a good performance for the future.
We thank the reviewer for pointing at this and we have corrected the text accordingly (line 527 in the marked-up version).

---

## Author Response (AR3)

**Executive editor**

We would like to thank the Executive editor for their feedback. Please find our responses (in blue) to the main points raised (shown in black) below.

I noticed that some of the figures are quite hard to read, particularly Figures 5, 8, 9 (primarely the vertical axis). I am therefore requesting that the readability of the figures is enhanced before the paper can be published.

In general, we increased the figures in size. We modified and replaced figures 5, 8 and 9 with larger font for y-axis labels and we increased them in resolution. We also introduced the sentence "For interpretation of the references to colour in this figure legend, the reader is referred to the web version of this article." to point any readers to the online version for correct colour interpretation and improved readability.

---

## Author Response (AR4)

**Executive editor**

We would like to thank the Executive editor for their feedback. Please find our responses (in blue) to the main points raised (shown in black) below.

Dear authors,

thanks for your effort regarding the readability of the figures. I think it is a bit better, but I'd leave possible further considerations for the technical editor.

From my side, the paper is now accepted for publication.

thanks for submitting your original work to NHESS

best regards,
Joaquim Pinto

Thank you.

Regards